# Development of Fertility, Social Status, and Social Trust of Farmers

**DOI:** 10.3390/ijerph19084759

**Published:** 2022-04-14

**Authors:** Liqing Li, He Jiang

**Affiliations:** School of Public Administration and Law, Hunan Agricultural University, Changsha 410128, China; liliqing1136@163.com

**Keywords:** children quantity of farmers, economic value, social value, emotional value, social status, social trust

## Abstract

Fertility, social status, and social trust are main social choice behaviors of Chinese farmers. This paper adopts the childbearing–value logic to establish a theoretical model of farmers’ childbearing–social status–social trust choices to examine the influence of farmers’ childbearing and social status on farmers’ social trust. The theoretical model showed that farmers will rationally choose the number of children to bear, emotional value, social value, economic value, social status, and social trust. The fertility of farmers’ children is actually a trade-off between quantity and value, and the fertility behavior affects social status through the direct mechanism of the number of children and the value of the adjustment mechanism, and together with the social status, through the direct mechanism, the adjustment mechanism of the number of children, the intermediate mechanism of social status, and the mixed adjustment mechanism. Asymmetry affects social trust equilibrium. Empirical research based on the CFPS (China Family Panel Studies) data in 2018 showed that farmers’ children quantity primarily inhibits, through the adjustment mechanism of children’s value–social status, social status and social trust; it exerts no direct impact or mediating effect on the social status. The economic value of children does not affect the social status, but it affects social trust through a positive child quantity adjustment mechanism, a negative social status mediation mechanism, and a negative mixed mediation mechanism. The social value of children affects social trust by the positive direct mechanism and the negative children quantity adjustment mechanism, as well as social trust by the negative direct mechanism, children quantity adjustment mechanism, children quantity–social status mixed adjustment mediating mechanism, and the positive social status–mediated mechanism. The emotional value of children affects the social status through the positive direct mechanism, as well as social trust through the positive direct mechanism, social status–mediated mechanism, and negative child quantity adjustment mechanism, and negative mixed mediation mechanism. Furthermore, social status positively impacts social trust rather than a symmetric transmission of the mediating effect of children’s value and the quantity adjustment effect of children’s value. However, no mediating effect of social trust was observed on children quantity. Social development leads to structural changes in the fertility value of farmers’ children, which makes farmers prefer their children’s social and economic value, exerting a complex impact on their own social status and social trust.

## 1. Introduction

Rapid industrialization, marketization, and urbanization of China have generally decreased farmers’ fertility and increased children’s value preference. Meanwhile, farmers’, social status is developing in the low class, and their social competition anxiety is amassing, resulting in a serious social trust crisis [1,2,3,4,5,6,7,8,9,10]. Thus, farmers’ selections of fertility, social status, social trust development, and how farmer fertility can affect farmer social and social trust issues warrant extensive research.

How do farmers make choices about childbearing, social status, and social trust? How does the birth of farmers’ children affect social status and social trust? How does social status affect social trust? Although many studies recognized the economic value, social value, and emotional value of peasants’ childbearing decisions [11,12,13,14,15,16,17], they focused on economics to establish a childbearing consumption model [18,19,20], a transaction model, or a consumption–investment model [21,22,23], and discuss people’s economic value incentives. In contrast, from the perspective of sociology and psychology, it explored the correlation between the social value, emotional value, and the quantity–quality or gender selection of people’s childbearing decisions [8,24,25,26,27,28,29,30]. The complex game between status and social trust and its constraints [4,26,27,31,32,33] was established, but no unified behavioral choice framework was adopted to analyze the balanced choices, mechanisms, and effects of farmers’ childbearing decisions, social status, and social trust. To make up for the insufficiency of existing research, this study first establishes a theoretical model of farmers’ children’s fertility–social status–social trust choices based on the existing new economics–family economics–children’s quantity–quality model, using the number–value logic of children’s birth, and highlights that farmers’ rationality, the number of children, economic value of children, social value of children, emotional value of children, social status, social trust, and balance of personal intertemporal consumption were selected appropriately, and that the number of children and the value of children affect society through asymmetric direct mechanisms, intermediary mechanisms, and adjustment mechanisms. Status and social trust not only affect social trust through a positive direct mechanism but also affect social trust through the mediating effect of the social trust effect of the number of children and the value of children. Second, using the empirical research to further elucidate the mechanism and effect of farmers’ children’s fertility, social status, and social trust, the specific findings are as follows: (1) The number of children does not directly affect the social status, but it inhibits the growth of farmers’ social status by adjusting the correlation between children’s social value and social status. The economic value of children exerts no direct impact on the social status of farmers, and the number of children regulates the effect. The social value and emotional value of children have no effect on the social status of farmers. The social value of children exerts a negative adjustment effect on the number of children on the social status of farmers. The fertility of farmers’ children primarily affects social status through social value and emotional value. (2) The direct mechanism of the economic value of farmers’ children does not affect social trust, but the social trust is improved through the adjustment mechanism of the number of children and the intermediary mechanism of social status, and the mixed adjustment mechanism of the number of children, and social status decreases social trust. (3) The social value of children decreases farmers’ social trust through the direct mechanism, adjustment mechanism of the number of children, and mixed adjustment mechanism of the number of children and social status, and improves social trust through the intermediary mechanism of social status. (4) The direct mechanism of children’s emotional value and the intermediary mechanism of social status improve farmers’ social trust, but the intermediary mechanism through the adjustment mechanism of the number of children and the mixed adjustment mechanism of the number of children and social status reduces the social trust of farmers. (5) The number of children exerts no significant effect on the social trust of farmers. In addition, empirical research found that the number of farmers’ children and the value of their children, together with farmers’ personality factors and family factors, affect social trust through the mediating mechanism of social status and the mixed mediation mechanism of the number of children and social status.

Based on the existing research, this study mainly makes the following contributions. First, it establishes a model of childbearing–social status–social trust selection, adopts the perspective of childbearing quantity–value, and systematically analyzes the number, economic value, social value, and emotion of farmers’ children. Value, social status, and social trust established an equilibrium, which is consistent with Becker [11], Easterlin [12], Schultz [13], Becker and Lewis [18], Liu and Lu [21], Guo and Gong [20], and Yuan [23], and other mainstream family economists have significantly different perspectives on the quantity and quality of childbearing. The value competition between childbearing, social status, and social trust is a game of value; thus, different childbearing balances are obtained. Second, the childbearing–social status–social trust selection model can explain the quantity–quality selection behavior of children as proposed by many mainstream family economists, such as Becker [11], Guo and Gong [20], and Yuan [23]. As the choice of the quantity–quality of children’s birth discussed by these scholars is actually equivalent to the choice of the number of children giving birth–economic value in this study, it can also make explanations consistent with the studies of Nauck [25], Shiue [27], Shen and Zhang [8], Gu [29], and Utomo et al. [30] in sociology and psychology. Third, it theoretically discovers the competition and substitution relationship, mechanism, and effect between farmers’ children’s fertility, social status, and social trust, and use empirical technology to identify and confirm the findings. Together with social status, it affects social trust, not just simple fertility–quality consumption [20], economic transactions [21], or the consumption–investment balance [23]. This has special theoretical inspiration for understanding the relationship and evolution of farmers’ fertility behavior, social status mobility, and social trust in the process of China’s modernization. Furthermore, this study provides implications for the development of mobility and social trust, presenting valuable theoretical and policy perspectives.

The remainder of this study is arranged as follows: Section 2 presents a literature review; Section 3 builds a theoretical model of social choice, analyzes the equilibrium choice of farmers’ children’s fertility, social status, and social trust and their connection mechanisms, and presents theoretical hypotheses; Section 4 empirically tests the theoretical hypotheses; and Section 5 presents the conclusions.

## 2. Literature Review

Many studies in economics, sociology, and psychology claimed that fertility has economic value, emotional value, and social value [11,12,13,14,15,16,17]. People often make rational fertility decisions based on their children’s economic, social, and emotional value. Becker et al. claimed that fertility is, indeed, the consumer or investment behavior of parents made in pursuit of their children’s economic value and that the choice of children quantity is balanced with variations in children’s economic value [11,12,13]. Nevertheless, people normally balance the choice of children quantity and quality to maximize economic utility and exhibit different children quantity–quality preferences, based on the economic ability and the correlation between the marginal cost and the marginal value of children quantity and quality [18]. The children quantity–quality consumption model indicates a substitution relationship between people’s children quantity consumption and children quality consumption as defined by education, and that an increase in children quality reduces fertility quantity [19,20]. The children consumption transaction model demonstrates that people balance children education investment, children wage compensation, and children’s economic value, and eventually choose the next best children education investment, and that children quality is in moderate equilibrium [21,22]. Yuan et al. used a children consumption–investment value model to endogenize children quantity, children quality, and family intergenerational transfer income and reported that the cost–income relationship of children quantity–quality reinforces public preference for children quality [23]. Briefly, in the new family economics paradigm, many scholars believe that the correlation between economic cost and economic value determines the balance of the number and quality of children to be born and that factors such as market development, family planning policies, land policies, changes in agricultural taxation, social endowment insurance, and family and personal abilities will affect farmers’ children. The correlation between fertility cost and economic value has a significant impact on the choice of farmers’ children’s fertility quantity and quality [4,10,28,34,35,36,37]. Gu et al. proposed that the emotional value of fertility comprises psychological satisfaction of raising children, marital happiness, and joy of child interaction, while social value comprises social status, social trust, clan power, marital stability, and family succession. As children’s social and emotional values differ, fertility decisions are different [25,29]. Hu and Chiang studied Taiwan, China, and reported that people with different children’s social values and emotional values had different fertility-selecting behaviors [38]. Kana et al. reported that Asian societies usually have a social value preference for sons, resulting in a preference for investing in sons rather than daughters [39,40]. Nanck claimed that the more people value children’s social value, the stronger the desire to have children early and a greater number of children, and the greater the children quantity; when people value children’s emotional value, they tend to have children later and have less children [25]. Nevertheless, no thorough theoretical discussion is available on how people’s economic, social, and emotional value joint preferences for children change regarding child quantity, and how people’s joint preferences for children’s value affect their fertility decisions.

People with different social statuses have different preferences for children’s value and different fertility choices. The urban middle class in Indonesia, Japan, and Turkey value their children’s emotional value most and focus less on social value; conversely, the rural class emphasizes their children’s economic value and focuses less on emotional value and social value, and the lower urban class values their children’s economic value as well as their children’s emotional and social value [31,32,41]. Shiue claimed that people with low social status, such as ordinary farmers, have a strong preference for quantity and have more children to have “more children and more happiness”; the middle class focuses on raising fewer but better children to attain the social aspiration of their children becoming outstanding people, and the social elites have sufficient economic resources and can afford to satisfy their child quantity and quality preference, and thereby raise more and better children to attain more children’s value [27]. Some studies claimed that there is no stable social status boundary for people’s fertility choices [4,42]. Reportedly, an increase in female economic status leads to a children quality preference and, thus, children quality increases, whereas an increase in male economic status increases the children quantity preference and, thus, children quantity increases [43]. Numerous empirical studies in the United States and Europe have illustrated that male economic status positively correlates with fertility quantity, whereas no definite relationship exists between female economic status and children quantity [44,45]. Research in China and other developing countries has revealed that children of different genders have different economic, social, and emotional values, especially the social value and economic value advantage of males “inheriting the lineage,” and peasant women improve their social status by giving birth to boys as a “mother is more expensive than the child” [24,26]. Utono et al. reported that when fertility reached a replacement level, the negative correlation between social status and children quantity started weakening or even disappeared [30]. Consequently, people’s fertility quantity follows a “U”-shaped distribution structure with social economic status; that is, the quantity of children is higher among the social elite and the lowest-class farmers, and lower among the middle class [46]. Nonetheless, the influence mechanism and effect of farmers’ children’s value and children quantity on social status remain partially discussed. 

With the comprehensive and systematic development of China’s economic and social modernization, the rural social structure has undergone incredible changes [47,48], the social class game has gradually weakened farmers’ social status acquisition ability, farmers’ social status is overall lowered and continuously differentiated, and farmers’ social status competition is under immense pressure [49,50]. In addition, farmers’ social status and social trust development face deeper social relationship constraints, social status mobility and inheritance are biased by various factors, and low-class social status solidification and paternal dependence are contrary to the free development of social trust [51]. To decrease social status competition pressure, people tend to reinforce their children’s social value utilization, social trust preference trust in blood relatives, and same class development, further locking class mobility, which could deepen social class differentiation, strengthen class status competition conflict [52], and markedly decrease the social trust level [53]. Shen et al. indicated that social status perceptions in China are polarized, with higher and lower social status groups having markedly lower levels of social trust, and that social status differentiation results in complex changes in social trust [8]. Fukuyama China is a low-trust society, where personal trust is based on blood order, especially farmers’ social trust [54], and thus farmers can easily fall into a vicious circle of low social status–low social trust, which has a strong intergenerational continuity [33]. These studies suggested that the mechanisms underlying the evolution of social trust driven by farmers’ social status in China are quite multifaceted; however, the specific mechanisms and effects remain unclear. 

To date, many studies have reported that the fertility of people in a particular social class is only a way to attain economic, social, and psycho-emotional satisfaction for themselves, and a competitive choice exists between social status and social trust. Stulp et al. claimed that a value substitution relationship exists between social status and fertility, determined by comparative advantage [55,56]. Indeed, conflicting resource allocations are present concerning people’s fertility and social status, and the interactive choice between social status and fertility is intricate [44,45]. Luo et al. reported that farmers’ children quantity positively correlated with the economic status in China; the richer the farmers are, the more children they give birth to, and the richer they become [4]. Shi reported that farmers’ economic status negatively correlated with children quantity in Hubei; the poorer the farmers are, the more children they give birth to, and the poorer they become. Having more children markedly weakened farmers’ economic status, having more children might not bring more happiness, and farmers might not seek happiness from their children [57]. Thus, a diverse combination relationship of farmers’ social status and fertility choice exists. Some sociologists reported that social trust is also an input–output process with economic, social, and emotional value and can increase people’s welfare level [58,59]. Nonetheless, the optimal choice of fertility, social status, and social trust merits further investigation.

With the expanding development of China’s economic and social reform, farmers’ survival and development budget tightened, the comprehensive costs of fertility grew rapidly, and the farmer fertility reduced rapidly to a worrying and extremely low level [48]. Reportedly, the limited quantitative choice space of farmer fertility in turn reinforces children quality preferences [60,61], while the marginal value of farmers’ children quality development is limited, and the instrumental value in social competition of fertility is diminished. Meanwhile, the competition value of farmers’ social status and social trust increased markedly, with a strong incentive to mobilize family resources for social status competition and social trust development. Hence, farmers carry out intrafamily cooperation and competition, which produces reciprocity and promotes internal trust, and competition results in intergenerational exploitation and conflict and damages internal trust [62], while internal intergenerational exploitation and conflict affect the intergenerational social status distribution of family members and seriously weaken intrafamily trust relationships [63], and even intensify them to an unbearable degree, resulting in brothers’ feuds and conflicts between husband and wife [5,64]. In contrast, external family cooperation and competition expand social trust, while competition conflict undermines social status and weakens farmers’ social status, solidifying their lower-class position [65], resulting in a severe social trust crisis. Despite the farmers’ best efforts to compete, the overall predicament of social competition has not changed fundamentally. Theoretically and realistically, fertility becomes the elementary choice for farmers to break their predicaments of social status competition and social trust development [4,61,66], which result in an intense farmer–government game of “over-birth” and “fine”; combined with the impact of market competition, farmers’ reproductive costs increased markedly, children quality decreased, and income decreased [6,67,68], thereby lowering the children’s value, which certainly strengthens farmers’ social status and social trust preference. However, still no theoretical explanation exists for the exact mechanism and effect. Although the Chinese government has relaxed birth control and implemented the three-child policy, there is more room for farmers to use fertility to influence their social status and social trust development, but how will fertility, social status, and social trust development be selected under the market mechanism? What is the transmission mechanism? How big is the effect? For questions such as these, theoretical answers are urgently needed. 

Thus, based on the existing analysis of children quantity–quality, this study uses fertility quantity–value logic to develop a selection model of farmer fertility–social status–social trust to determine the intrinsic mechanism and action effect of fertility on farmers’ social status and social trust, and uses an econometric model to determine the action mechanism and influence effect among farmer fertility, social status, and social trust. A special theoretical inspiration exists for understanding the correlation among farmer fertility, social status mobility, and social trust in China, and provides a useful theoretical and policy perspective for comprehending the impact of major changes in national family planning on farmer fertility, social mobility, and social trust development.

## 3. Theoretical Models

### 3.1. Hypotheses

Farmer family fertility follows a quantity–quality consumption model [9] and children consumption–investment model; however, most of these theoretical models focus on personal children quantity–quality consumption or intergenerational exchange economic value [14], overlooking children’s social value and emotional value, and lacking social status and social trust choices. Thus, by combining the two models mentioned above [14], a model of farmers’ consumption, fertility, social status, and social trust choices was developed, focusing on the analysis of the endogenous mechanisms and utility equilibria of farmers’ fertility, social status, and social trust.

To simplify the analysis, we assume that rural society comprises numerous homogeneous farmers. The life of a farmer is divided into two periods: young and old. The number of young farmers in period *t* is *L_t_*, and if the rural population growth is *n_t_*, the number of farmers in period *t* + 1 is $L_{t+1}=\left(1+n_{t}\right)L_{t}$. The farmer is employed in his or her youth, and income *w_t_* from employment is used for social status, social trust development, fertility, personal consumption, savings, and taxes. In old age, farmers simply use personal savings with interest, intergenerational transfer income from children, and government pensions for consumption. 

Farmer fertility has economic, social, and emotional value returns. The children’s economic value is primarily the transfer income from child support in the farmer’s old age; the more the farmer receives from his/her children, the higher the children’s economic value. In rural China, it is common to use the “shared support” or “rotating support” approach to equally undertake maintenance obligations [69], assuming that the support economic value received from each child by a farmer in old age is $w_{z}$, and the total economic value received from $n_{t}$ children is $n_{t}w_{z}$. The children’s social value denotes the social reputation, social trust, inheritance, and other social satisfaction values brought by children social status. The Chinese cultural tradition of continuing the ancestral line gives children an extremely crucial social value of succession. The more “successful” the children are, the more respectable the farmer feels in social interactions, the higher their social reputation in the village, and the higher the children’s social value [60]. Assuming that the children quantity $n_{t}$ gives the social value of succession, $e_{t}$ gives the social value of the single child, and $n_{t}e_{t}$ is the total social value of farmer fertility. Emotional value denotes the emotional trust in the child-rearing process, the joy of interaction and the care of children in old age [16]; the greater the number of children, the better the relationship with children, i.e., “more children, more happiness,” and the greater the children’s emotional value [48]. 

A rational farmer will choose consumption $c_{1t}$ at young age, consumption $c_{2t+1}$ in old age, social status $d_{1t}$, social trust $R_{1t}$, children quantity $n_{t}$, children’s emotional value $q_{t}$, children’s social value $e_{t}$, and economic value $w_{z}$ based on income constraints, maximizing the individual utility. The farmer utility function is assumed to be $U_{t}$: (1)$$U_{t}=a_{1}\ln c_{1t}+a_{2}\ln d_{1t}+a_{3}\ln R_{1t}+a_{4}\ln c_{2t+1}+\beta_{1}\ln(n_{t}w_{z})+\beta_{2}\ln(n_{t}e_{t})+\beta_{3}\ln(n_{t}q_{t})+\beta_{4}\ln(n_{t})$$
where $a_{1},\;a_{2},a_{3},a_{4}$ denote the utility contribution coefficients of the farmer’s young-age consumption $c_{1t}$, social status $d_{t}$, social trust $R_{1t}$, and old-age consumption $c_{2t+1}$; $\beta_{1},\beta_{2},\beta_{3},\beta_{4}$ denote the utility contribution coefficients of children’s economic value, social value, emotional value, and children quantity; $a_{1}\ln c_{1t}$, $a_{2}\ln d_{1t}$, $a_{3}\ln R_{1t}$, and $a_{4}\ln c_{2t+1}$ are the farmer’s young-age consumption utility, social status utility, social trust utility, and old-age consumption utility, respectively; and $\beta_{1}\ln(n_{t}w_{z})$, $\beta_{2}\ln(n_{t}e_{t})$, $\beta_{3}\ln(n_{t}q_{t})$, and $\beta_{4}\ln(n_{t})$ denote the utility of children’s economic value, social value, emotional value, and children quantity, respectively. 

The farmer’s consumption, social status, social trust, and fertility have costs. Assuming that the prices of the farmer’s young- and old-age consumer goods are $p_{{}_{1t}}$ and $p_{{}_{2t+1}}$, the total cost of consumption is $p_{{}_{1t}}c_{1t}$ and $p_{2t+1}c_{2t+1}$, respectively. The marginal costs of social status and social trust development are $p_{dt}$ and $p_{\mathrm{Rt}}$, respectively, while the total cost is $p_{dt}d_{1t}$ and $p_{\mathrm{Rt}}R_{1t}$, respectively. The farmer fertility costs comprise the following components. The first part is the economic value cost. The marginal cost of the economic value of the farmer’s single child is $p_{wt}$. The total cost of the $n_{t}$ children’s economic value is $p_{wt}n_{t}w_{z}$. Clearly, the more economic value the farmer obtains from children, the greater the cost. The second part is the children’s social value cost. Currently, it is common for both citizens and farmers to support their children’s “house and car” and “education” to enhance their social status and competition power in the marriage market [70]. Experience demonstrates that the higher the social value of farmers’ children, the higher the number of children, and the higher the cost of obtaining children’s social value [71]. Assuming that the marginal cost of the farmer’s single-child social value is $p_{ne}$, then the total cost of the social value input for $n_{t}$ children is $p_{ne}n_{t}e_{t}$. The third part is the cost of the children’s emotional value. When the marginal cost of single-child emotion is $p_{qt}$, the total cost for the farmer obtaining the children’s emotional value is $p_{qt}n_{t}q_{t}$. The fourth part is the cost of “over-birth.” The farmer faces huge fines for “over-birth” in violation of the family planning policy, and even other costs, such as the loss of property and job opportunities owing to “over-birth” [48,57]. If the farmers’ children quantity, $n_{t}$, exceeds the national legal number of children, $n_{0}$, an “over-birth” fee is paid. Generally, the more farmer “over-births,” the higher the “over-birth” fee, which is usually several times the farmer’s per capita income, assuming that the total number of various “over-birth” fees is $m(n_{t}-n_{0})w_{t}$. Herein, $w_{t}$ is the farmer’s wage income, and *m* > 0 is the multiplier of the over-birth penalty; the larger *m* is, the smaller $n_{0}$ is, and the more severe the control of rural family planning. 

Considering that the government levies a one-time tax on farmers’ labor income at the rate of $t_{g}$, the government gives a single farmer’s child a support allowance of $y_{et}$ and gives a farmer old-age security of $y_{lt}$, for which $r_{t}$ is the interest and $S_{t}$ is the farmer’s youth savings. Then, the farmer’s lifetime budget constraint is: (2)$$\begin{array}{cc} & p_{1t}c_{1t}+p_{dt}d_{1t}+p_{Rt}R_{1t} \end{array}+p_{ne}n_{t}e_{t}+p_{zt}n_{t}w_{z}+p_{qt}n_{t}q_{t}+S_{t}\leq [1-t_{g}-m(n_{t}-n_{0})]w_{t}+n_{t}e_{t}y_{et}$$
(3)$$\begin{array}{cc} & \begin{array}{cc} & p_{2t+1}c_{2t+1}=(1+r_{t} \end{array} \end{array})S_{t}+y_{lt}+n_{t}w_{z}$$

Assuming that the production function is $Y_{t}=F\left(K_{t},A_{t}L_{t}\right)$, with the constant scale remuneration and satisfying the paddy field condition, the rural labor and capital markets are perfectly competitive, labor and capital compete to attain their marginal output, and there is no depreciation. Then, the firm’s profit maximizing wage level *W_t_* paid to the young farmer is as follows: (4)$$w_{t}=f\left(k_{t}\right)-f^{\prime }\left(k_{t}\right)k_{t}$$
where *k_t_* is the unit labor capital. 

The government is assumed to levy a one-time tax on labor income, while providing children education support, old-age security, and other public services to farmer households, with the remainder going to government savings. 

Thus, the government budget dynamic equation is as follows: (5)$$S_{gt+1}=\left(1+r_{t}\right)S_{gt}+t_{g}L_{t}w_{t}+mL_{t}(n_{t}-n_{0})w_{t}-L_{t}n_{t}e_{t}y_{et}-L_{t}y_{lt}-G_{t}$$
where $S_{g}{}_{t}$ is the government savings in period *t*, $S_{g}{}_{t+1}$ denotes the government savings in period *t*+1, $t_{g}L_{t}w_{t}$ denotes the government one-time labor tax, $mL_{t}{(n}_{t}{-n}_{0}{)w}_{t}$ denotes the government “over-birth” penalty income, $L_{t}n_{t}e_{t}y_{et}$ is the government children support subsidy, $L_{t}y_{lt}$ is the old-age security expenditure, and *G_t_* is other government public expenditure in the period *t*. When the government savings are in dynamic equilibrium, then:$\frac{S_{gt+1}}{L_{t+1}}=\frac{S_{gt}}{L_{t}}=s_{g}$; when the per capita government budget is in equilibrium, then: (6)$$\left(1+r_{t}-n_{t}\right)s_{gt}+[t_{g}+m(n_{t}-n_{0})]w_{t}-n_{t}e_{t}y_{et}-y_{lt}-g_{t}=0$$
where *s_gt_* denotes government per capita savings and *g_t_* is government per capita expenditure on other public services. 

Based on the previous assumptions, the farmer’s individual behavior optimization problem is as follows: (7)$$\begin{array}{l} MaxU_{t}(c_{1t},d_{1t},R_{1t},c_{2t+1},n_{t},e_{t},q_{t},w_{t})=a_{1}\ln c_{1t}+a_{2}\ln d_{1t}+a_{3}\ln R_{1t}+a_{4}\ln c_{2t+1}\\ +\beta_{1}\ln(n_{t}w_{z})+\beta_{2}\ln(n_{t}e_{t})+\beta_{3}\ln(n_{t}q_{t})+\beta_{4}\ln(n_{t}) \end{array}\;s.t\;p_{1t}c_{1t}+p_{dt}d_{1t}+p_{Rt}R_{1t}+p_{ne}n_{t}e_{t}+p_{qt}n_{t}q_{t}+p_{wt}n_{t}w_{z}+S_{t}\leq [1-t_{g}-m(n_{t}-n_{0})]w_{t}+n_{t}e_{t}y_{et}\;p_{2t+1}c_{2t+1}=(1+r_{t})S_{t}+y_{lt}+n_{t}w_{z}\;\left(1+r_{t}-n_{t}\right)s_{gt}+[t_{g}+m(n_{t}-n_{0})]w_{t}-n_{t}e_{t}y_{et}-y_{lt}-g_{t}=0$$

### 3.2. Farmer Optimal Selection Equilibrium

#### Farmers’ Fertility, Social Status, and Social Trust Choice Equilibrium

By solving Equation (7), it can be deduced that farmer optimal consumption, social status, social trust, children quantity, children’s economic value, social value, emotional value, and children quantity are in equilibrium: (8)$$c_{1t}^{*}=\frac{a_{1}w_{t}^{\prime }}{(1+r_{t})\left(\sum_{i=1}^{4}a_{i}+\sum_{j=1}^{3}\beta_{j}+f(\theta )\sum_{j=1}^{4}\beta_{j}\right)p_{1t}}$$
(9)$$d_{1t}^{*}=\frac{a_{2}w_{t}^{\prime }}{(1+r_{t})\left(\sum_{i=1}^{4}a_{i}+\sum_{j=1}^{3}\beta_{j}+f(\theta )\sum_{j=1}^{4}\beta_{j}\right)p_{dt}}$$
(10)$$R_{1t}^{*}=\frac{a_{3}w_{t}^{\prime }}{(1+r_{t})\left(\sum_{i=1}^{4}a_{i}+\sum_{j=1}^{3}\beta_{j}+f(\theta )\sum_{j=1}^{4}\beta_{j}\right)p_{Rt}}$$
(11)$$c_{2t+1}^{*}=\frac{a_{4}w_{t}^{\prime }}{\left(\sum_{i=1}^{4}a_{i}+\sum_{j=1}^{3}\beta_{j}+f(\theta )\sum_{j=1}^{4}\beta_{j}\right)p_{2t+1}}$$
(12)$$w_{z}^{*}=\frac{\beta_{1}\left(r_{t}mw_{t}+s_{gt}\right)}{f(\theta )(\beta_{1}+\beta_{2}+\beta_{3})\left((1+r_{t})p_{wz}-1\right)}$$
(13)$$e_{t}^{*}=\frac{\beta_{2}\left(r_{t}mw_{t}+s_{gt}\right)}{f(\theta )(\beta_{1}+\beta_{2}+\beta_{3})\left((1+r_{t})p_{ne}-r_{t}y_{et}\right)}$$
(14)$$q_{t}^{*}=\frac{\beta_{3}\left(r_{t}mw_{t}+s_{gt}\right)}{(1+r_{t})f(\theta )\left(\beta_{1}+\beta_{2}+\beta_{3}\right)p_{qt}}$$
(15)$$n_{t}^{*}=\frac{f(\theta )\left(\beta_{1}+\beta_{2}+\beta_{3}\right)w_{t}^{\prime }}{\left(\sum_{i=1}^{4}a_{i}+\sum_{j=1}^{3}\beta_{j}+f(\theta )\sum_{j=1}^{4}\beta_{j}\right)\left(r_{t}mw_{t}+s_{gt}\right)}$$
where $f(\theta )=\frac{\theta_{1}\theta_{2}\theta_{3}-2-\theta_{1}-\theta_{2}-\theta_{3}}{4+2\theta_{1}+2\theta_{2}+2\theta_{3}+\theta_{1}\theta_{2}+\theta_{1}\theta_{3}+\theta_{2}\theta_{3}}$, $\theta_{1}=\frac{\beta_{2}+\beta_{3}+\beta_{4}}{\beta_{1}}$, $\theta_{2}=\frac{\beta_{1}+\beta_{3}+\beta_{4}}{\beta_{2}}$, $\theta_{3}=\frac{\beta_{1}+\beta_{2}+\beta_{4}}{\beta_{3}}$, $w_{t}^{\prime }=(1+r_{t}+r_{t}mn_{0}-r_{t}t_{g})w_{t}+(1+r_{t})s_{gt}-g_{t}$. 

Under the asymmetric constraint of various factors, farmers rationally choose the social status, social trust, children quantity, economic value, social value, and emotional value equilibrium. 

### 3.3. Effect of Fertility on Social Status and Social Trust

#### 3.3.1. Direct Effect of Children’s Value on Farmer Social Trust

The asymmetry of farmers’ children’s economic value, social value, and emotional value affects farmer social trust. Through the optimization solution, the response functions of farmers’ social trust to children’s economic value, social value, and emotional value are $R_{1t}^{*}=\frac{a_{3}\left[(1+r_{t})p_{wt}-1\right]n_{t}^{*}w_{z}^{*}}{\beta_{3}(1+r_{t})p_{Rt}}$, $R_{1t}^{*}=\frac{a_{3}\left[(1+r_{t})p_{ne}-r_{t}y_{et}\right]n_{t}^{*}e_{t}^{*}}{\beta_{2}(1+r_{t})p_{Rt}}$, and $R_{1t}^{*}=\frac{a_{3}p_{qt}n_{t}^{*}q_{t}^{*}}{\beta_{1}p_{Rt}}$. Thus, the direct mechanism effects of the economic value, social value, and emotional value of farmers’ children are as follows:$\frac{\partial R_{1t}^{*}}{\partial w_{z}^{*}}=\frac{a_{3}\left[(1+r_{t})p_{wt}-1\right]n_{t}^{*}}{\beta_{3}(1+r_{t})p_{Rt}}$, $\frac{\partial R_{1t}^{*}}{\partial e_{t}^{*}}=\frac{a_{3}\left[(1+r_{t})p_{ne}-r_{t}y_{et}\right]n_{t}^{*}}{\beta_{2}(1+r_{t})p_{Rt}}$, and $\frac{\partial R_{1t}^{*}}{\partial q_{t}^{*}}=\frac{a_{3}p_{qt}n_{t}^{*}}{\beta_{1}p_{Rt}}$; therefore, farmer social trust increases with children’s economic value at $(1+r_{t})p_{wt}>1$, increases with children’s social value at $(1+r_{t})p_{ne}>r_{t}y_{et}$, and increases unconditionally with children’s emotional value. Hence, the following hypothesis is proposed:

**Hypothesis** **1** **(H1).**
*(1) Children’s emotional value exerts a positive direct impact on farmers’ social trust; (2) when*

$(1+r_{t})p_{wt}>1$

*, economic value exerts a positive direct impact on farmers’ social trust; (3) when*

$(1+r_{t})p_{ne}>r_{t}y_{et}$

*, the social value of children exerts a positive direct impact on the social trust of farmers.*


Children’s emotional value exerts a direct positive impact on farmers’ social trust, and children’s social and economic value have an uncertain effect on farmer social trust.

#### 3.3.2. Direct Impact of Children’s Value on Farmers’ Social Status

The response functions of farmers’ social status to children’s economic value, social value, and emotional value can be obtained from the farmer behavior optimization equation, and the response functions are $d_{1t}^{*}=\frac{a_{2}\left[(1+r_{t})p_{wt}-1\right]n_{t}^{*}w_{z}^{*}}{\beta_{3}(1+r_{t})p_{dt}}$, $d_{1t}^{*}=\frac{a_{2}\left[(1+r_{t})p_{ne}-r_{t}y_{et}\right]n_{t}^{*}e_{t}^{*}}{\beta_{2}(1+r_{t})p_{dt}}$, and $d_{1t}^{*}=\frac{a_{2}p_{qt}n_{t}^{*}q_{t}^{*}}{\beta_{1}p_{dt}}$, respectively, to obtain $\frac{\partial d_{1t}^{*}}{\partial w_{z}^{*}}>0$ at $(1+r_{t})p_{wt}>1$, $\frac{\partial d_{1t}^{*}}{\partial e_{t}^{*}}>0$ at $(1+r_{t})p_{ne}>r_{t}y_{et}$, $\frac{\partial d_{1t}^{*}}{\partial q_{t}^{*}}>0$. The increase in children’s emotional value will improve farmers’ social status, but the increase in farmers’ social value and economic value may not improve farmers’ social status. Hence, the following hypothesis is proposed: 

**Hypothesis** **2** **(H2).**
*(1) Children’s emotional value has a positive direct impact on farmers’ social status; (2) when*

$(1+r_{t})p_{wt}>1$

*, economic value exerts a positive direct impact on farmers’ social status; (3) when*

$(1+r_{t})p_{ne}>r_{t}y_{et}$

*, the social value of children exerts a positive direct impact on the social status of farmers.*


#### 3.3.3. Impact of Farmers’ Social Status on Farmer Social Trust

Farmers’ social status has a complex effect on social trust. Per farmers’ social choice decisions, the response equation of farmers’ social trust to social status can be obtained as follows:$R_{1t}^{*}=\frac{a_{3}p_{dt}d_{1t}^{*}}{a_{2}p_{Rt}}=\frac{\left(a_{3}/p_{Rt}\right)d_{1t}^{*}}{\left(a_{2}/p_{dt}\right)}$. Farmers’ social trust is directly proportional to farmers’ social status, and the equilibrium development level of social trust and social status is determined per the comparative advantage relationship between social trust and social status, and is not affected by the value of children and the number of children. Hence, the following hypothesis is proposed:

**Hypothesis** **3** **(H3).**
*The growth of farmers’ social status exerts a direct positive impact on farmer social trust.*


#### 3.3.4. The Mediating Effect of Social Status of Children’s Value on Farmers’ Social Trust

As children’s value affects farmers’ social status, and farmers’ social status in turn affects farmer social trust, social status as a mediator of children’s value affects social trust, asymmetrically transmits the children’s value mediating effect, and then affects social trust. Combined with the previous analysis, the mediated response function of children’s economic value, social value, and emotional value mediated by social status influencing farmer social trust is as follows: (16)$$\begin{array}{c} R_{1t}^{*}=\frac{a_{3}p_{dt}d_{1t}^{*}}{a_{2}p_{Rt}}=\frac{a_{3}(1+r_{t})p_{qt}n_{t}^{*}q_{t}^{*}}{\left(\beta_{1}+\beta_{2}+\beta_{3}+\beta_{4}\right)p_{Rt}}+\frac{a_{3}\left((1+r_{t})p_{ne}-r_{t}y_{et}\right)n_{t}^{*}e_{t}^{*}}{\left(\beta_{1}+\beta_{2}+\beta_{3}+\beta_{4}\right)p_{Rt}}+\frac{a_{3}\left((1+r_{t})p_{wt}-1\right)n_{t}^{*}w_{z}^{*}}{\left(\beta_{1}+\beta_{2}+\beta_{3}+\beta_{4}\right)p_{Rt}}\\ +\frac{a_{3}\left(r_{t}mw_{t}+s_{gt}\right)n_{t}^{*}}{\left(\beta_{1}+\beta_{2}+\beta_{3}+\beta_{4}\right)p_{Rt}} \end{array}$$

In Equation (16), $\frac{a_{3}\left((1+r_{t})p_{wt}-1\right)n_{t}^{*}w_{z}^{*}}{\left(\beta_{1}+\beta_{2}+\beta_{3}+\beta_{4}\right)p_{Rt}}$ denotes the social status–mediated effect of children’s economic value, $\frac{a_{3}\left((1+r_{t})p_{ne}-r_{t}y_{et}\right)n_{t}^{*}e_{t}^{*}}{\left(\beta_{1}+\beta_{2}+\beta_{3}+\beta_{4}\right)p_{Rt}}$ denotes the social status–mediated effect of children’s social value, and $\frac{a_{3}(1+r_{t})p_{qt}n_{t}^{*}q_{t}^{*}}{\left(\beta_{1}+\beta_{2}+\beta_{3}+\beta_{4}\right)p_{Rt}}$ denotes the social status–mediated effect of children’s emotional value. The social trust mediating effects of children’s economic, social, and emotional values have the following basic characteristics: (i) The social status–mediated effects of children’s economic, social, and emotional values are asymmetric; (ii) the social status–mediated effects of children’s economic, social, and emotional values are asymmetrically constrained by each factor; and (iii) the social status–mediated effects of children’s value vary inconsistently. The children’s economic value mediating effect increases with economic value at $(1+r_{t})p_{wt}>1$, the children’s social value mediating effect increases with children’s social value at $(1+r_{t})p_{ne}>r_{t}y_{et}$, and the children’s emotional value mediating effect increases with the children equilibrium emotional value. Hence, the following hypothesis is proposed:

**Hypothesis** **4** **(H4).**
*(1) The economic value, social value, and emotional value of farmers’ children affect social trust through an asymmetric social status intermediary mechanism; (2) when*

$(1+r_{t})p_{wt}>1$

*, a child’s economic value growth has a positive social status mediating effect on social trust; (3) when*

$(1+r_{t})p_{ne}>r_{t}y_{et}$

*, a child’s social value growth has a positive social status mediating effect on social trust; and (4) the growth of children’s emotional value has a positive social status mediating effect on social trust.*


#### 3.3.5. Effect of Children Quantity on Farmers’ Children’s Value, Social Status, and Social Trust

Per the theoretical model, as one of farmer’s rational choices, children quantity asymmetrically affects social trust through complex mechanisms, as the response functions of children’s economic value, social value, and emotional value on the children quantity are: (17)$$(1+r_{t})\left(\beta_{2}+\beta_{3}+\beta_{4}\right)p_{qt}q_{t}^{*}=\beta_{1}\left((1+r_{t})p_{ne}-r_{t}y_{et}\right)e_{t}^{*}+\beta_{1}\left((1+r_{t})p_{wt}-1\right)w_{z}^{*}+\beta_{1}\left(r_{t}mw_{t}+s_{gt}\right)$$
(18)$$\left(\beta_{1}+\beta_{3}+\beta_{4}\right)\left((1+r_{t})p_{ne}-r_{t}y_{et}\right)e_{t}^{*}=\beta_{2}(1+r_{t})p_{qt}q_{t}^{*}+\beta_{2}\left((1+r_{t})p_{wt}-1\right)w_{z}^{*}+\beta_{2}\left(r_{t}mw_{t}+s_{gt}\right)$$
(19)$$\left(\beta_{1}+\beta_{2}+\beta_{4}\right)\left((1+r_{t})p_{wt}-1\right)w_{z}^{*}=\beta_{3}(1+r_{t})p_{qt}q_{t}^{*}+\beta_{3}\left((1+r_{t})p_{ne}-r_{t}y_{et}\right)e_{t}^{*}+\beta_{3}\left(r_{t}mw_{t}+s_{gt}\right)$$

Equations (17) and (18) are both independent of $n_{t}^{*}$, and the farmers’ children quantity exerts no direct impact on children’s economic value, social value, and emotional value. 

As the number of children does not directly affect the value of the children, the economic, social, and emotional values of children do not mediate farmers’ social status and social trust through the children quantity. Nonetheless, as the response functions of farmers’ social trust to children’s economic, social, and emotional values are $R_{1t}^{*}=\frac{a_{3}\left[(1+r_{t})p_{wt}-1\right]n_{t}^{*}w_{z}^{*}}{\beta_{3}(1+r_{t})p_{Rt}}$,$R_{1t}^{*}=\frac{a_{3}\left[(1+r_{t})p_{ne}-r_{t}y_{et}\right]n_{t}^{*}e_{t}^{*}}{\beta_{2}(1+r_{t})p_{Rt}}$,$R_{1t}^{*}=\frac{a_{3}p_{qt}n_{t}^{*}q_{t}^{*}}{\beta_{1}p_{Rt}}$, respectively, changes in children quantity will asymmetrically regulate the correlation between farmer social trust and children’s economic, social, and emotional values, which has the children’s value–social trust moderating effect. Likewise, as the response functions of farmers’ social status to children’s economic value, social value, and emotional value are $d_{1t}^{*}=\frac{a_{2}\left[(1+r_{t})p_{wt}-1\right]n_{t}^{*}w_{z}^{*}}{\beta_{3}(1+r_{t})p_{dt}}$, $d_{1t}^{*}=\frac{a_{2}\left[(1+r_{t})p_{ne}-r_{t}y_{et}\right]n_{t}^{*}e_{t}^{*}}{\beta_{2}(1+r_{t})p_{dt}}$, and $d_{1t}^{*}=\frac{a_{2}p_{qt}n_{t}^{*}q_{t}^{*}}{\beta_{1}p_{dt}}$, respectively, the children quantity asymmetrically regulates the correlation among children’s economic value, social value, emotional value, and farmers’ social status, for which the number of children also has a moderating effect. Moreover, social status and social trust response functions on children quantity are: (20)$$d_{1t}^{*}=\frac{a_{2}\left[(1+r_{t})p_{qt}q_{t}^{*}+\left((1+r_{t})p_{ne}-r_{t}y_{et}\right)e_{t}^{*}+\left((1+r_{t})p_{wt}-1\right)w_{z}^{*}+r_{t}mw_{t}+s_{gt}\right]n_{t}^{*}}{(1+r_{t})\left(\beta_{1}+\beta_{2}+\beta_{3}+\beta_{4}\right)p_{dt}}$$
(21)$$R_{1t}^{*}=\frac{a_{3}\left[(1+r_{t})p_{qt}q_{t}^{*}+\left((1+r_{t})p_{ne}-r_{t}y_{et}\right)e_{t}^{*}+\left((1+r_{t})p_{wt}-1\right)w_{z}^{*}+r_{t}mw_{t}+s_{gt}\right]n_{t}^{*}}{(1+r_{t})\left(\beta_{1}+\beta_{2}+\beta_{3}+\beta_{4}\right)p_{Rt}}$$

As long as $(1+r_{t})p_{qt}q_{t}^{*}+\left((1+r_{t})p_{ne}-r_{t}y_{et}\right)e_{t}^{*}+\left((1+r_{t})p_{wt}-1\right)w_{z}^{*}+r_{t}mw_{t}+s_{gt}\neq 0$, children quantity exerts a direct effect utility on farmers’ social status and social trust. Meanwhile, as social status directly affects social trust, children quantity affects social trust through social status–mediated mechanisms. Nonetheless, the response of farmers’ social trust to social status does not correlate with children quantity; thus, the children quantity does not moderate the correlation between social trust and social status and does not exert a moderating effect. Hence, the following hypothesis is proposed:

**Hypothesis** **5** **(H5).**
*(1) Children quantity exerts no direct impact on children’s economic value, social value, and emotional value; (2) children quantity exerts a direct asymmetric impact on social trust and social status at*

$(1+r_{t})p_{qt}q_{t}^{*}+\left((1+r_{t})p_{ne}-r_{t}y_{et}\right)e_{t}^{*}+\left((1+r_{t})p_{wt}-1\right)w_{z}^{*}+r_{t}mw_{t}+s_{gt}\neq 0$

*; (3) children quantity has asymmetric children’s economic value–social status, social value–social status, and emotional value–social status moderating effects; (4) children quantity has asymmetric children’s economic value–social trust, social value–social trust, and emotional value–social trust moderating effects, children’s value has no children quantity mediating mechanism affecting social trust; and (5) children quantity has no farmers’ social status–social trust moderating effect, but under certain conditions, there is a social status–mediated effect affecting social trust.*


#### 3.3.6. Mixed Moderating Mediating Effects of the Number of Children, Children’s Value and Social Status on Social Trust

As the economic value, social value, and emotional value of farmers’ children’s fertility affect social trust through the asymmetric social status intermediary mechanism and not through the intermediary mechanism of the number of children, and the number of children regulates the economic value–social status, social value–social status, and emotional value of children–social status, the economic value, social value, and emotional value of farmers’ children’s fertility affect social trust, but only the number of children–social status mixed mediation mechanism. Theoretically, the number of children does not regulate the social status–social trust relationship; thus, the value of children can only be adjusted through the mixed mediation mechanism of children’s value–social status. Specifically, the number of children and the value of children are cross-multiplied to affect social status, and affect social trust through social status intermediary transmission, rather than through the social status–social trust mixed mediation mechanism (the number of children regulates the social status–social status trust relationship) and children’s value–social status–social trust mixed mediation mechanism (the number of children simultaneously adjusts children’s value–social status, social status–social trust relationship) affects social trust. Per the theoretical model, the economic value, social value, and emotional value of farmers’ children’s fertility mainly affect social trust through the direct mechanism, the mediating mechanism of social status, the regulating mechanism of children’s value–social trust and the mixed regulating mechanism of children’s value–social status. According to the model equilibrium of $d_{1t}^{*}=\frac{a_{2}\left[(1+r_{t})p_{wt}-1\right]n_{t}^{*}w_{z}^{*}}{\beta_{3}(1+r_{t})p_{dt}}$,$d_{1t}^{*}=\frac{a_{2}\left[(1+r_{t})p_{ne}-r_{t}y_{et}\right]n_{t}^{*}e_{t}^{*}}{\beta_{2}(1+r_{t})p_{dt}}$, and $d_{1t}^{*}=\frac{a_{2}p_{qt}n_{t}^{*}q_{t}^{*}}{\beta_{1}p_{dt}}$, the moderating effects of the number of children regulating children’s value–social status are $\frac{\partial d_{1t}^{*}}{\partial (n_{t}^{*}w_{z}^{*})}=\frac{a_{2}\left[(1+r_{t})p_{wt}-1\right]}{\beta_{3}(1+r_{t})p_{dt}}$, $\frac{\partial d_{1t}^{*}}{\partial (n_{t}^{*}e_{t}^{*})}=\frac{a_{2}\left[(1+r_{t})p_{ne}-r_{t}y_{et}\right]}{\beta_{2}(1+r_{t})p_{dt}}$, and $\frac{\partial d_{1t}^{*}}{\partial (n_{t}^{*}q_{t}^{*})}=\frac{a_{2}p_{qt}}{\beta_{1}p_{dt}}$, respectively, while the social status directly affects the social trust response function of $R_{1t}^{*}=\frac{a_{3}p_{dt}d_{1t}^{*}}{a_{2}p_{Rt}}=\frac{\left(a_{3}/p_{Rt}\right)d_{1t}^{*}}{\left(a_{2}/p_{dt}\right)}$, so the mediating coefficient of social status, $\phi =\frac{\partial R_{1t}^{*}}{\partial d_{1t}^{*}}=\frac{a_{3}p_{dt}}{a_{2}p_{Rt}}$, is a constant that has nothing to do with the number of children born, the value of children, social status, or social trust. Therefore, the children’s value–social status mixed mediating effects of children’s economic value, social value, and emotional value are $\frac{\partial d_{1t}^{*}}{\partial (n_{t}^{*}w_{z}^{*})}\frac{\partial R_{1t}^{*}}{\partial d_{1t}^{*}}=\frac{a_{2}\phi \left[(1+r_{t})p_{wt}-1\right]}{\beta_{3}(1+r_{t})p_{dt}}$, $\frac{\partial d_{1t}^{*}}{\partial (n_{t}^{*}e_{t}^{*})}\frac{\partial R_{1t}^{*}}{\partial d_{1t}^{*}}=\frac{a_{2}\phi \left[(1+r_{t})p_{ne}-r_{t}y_{et}\right]}{\beta_{2}(1+r_{t})p_{dt}}$, and $\frac{\partial d_{1t}^{*}}{\partial (n_{t}^{*}q_{t}^{*})}\frac{\partial R_{1t}^{*}}{\partial d_{1t}^{*}}=\frac{\phi a_{2}p_{qt}}{\beta_{1}p_{dt}}$, respectively. From this, Hypothesis 6 can be deduced:

**Hypothesis** **6** **(H6).***(1) The value of peasant children affects social trust through the asymmetric value of children–social status mixed to regulate the value of intermediaries**; (2) when*$(1+r_{t})p_{wt}>1$*, children’s economic value increases, social trust is positive, and the children’s value–social status mix moderates the mediating effect**; (3) when*$(1+r_{t})p_{ne}>r_{t}y_{et}$*, the child’s social value increase has a positive child’s value–social status mixed mediating effect on social trust**; (4) the growth of children’s emotional value has a positive children’s value–social status mixed mediating effect on social trust*.

Hence, fertility economic value, social value, emotional value, and children quantity asymmetrically affect farmers’ social status and social trust through a complex direct acting mechanism, social status intermediary mechanism, child quantity adjustment mechanism, and children’s value–social status mixed mediation mechanism, whereas social status affects social trust through direct mechanisms and children’s value and quantity mediating mechanisms (Figure 1). 

## 4. Empirical Study

### 4.1. Empirical Strategy

To test the hypotheses mentioned above, based on OLS and logit methods, we constructed a moderating effect model, a mediating effect model, and a mixed moderation mediating effect model for the empirical analysis. The specific empirical strategy and steps are shown below: 

The first step was to confirm the mechanism and effect of farmer fertility economic value, social value, and emotional value on social trust, and test the partial content of theoretical Hypothesis 1 and Hypothesis 5 to ascertain the direct mechanisms, regulatory mechanisms, and the effects of farmer fertility on social trust. In addition, a moderating effect model was constructed (Figure 2), with the farmer fertility economic value, social value, and emotional value as the key explanatory variables, the children quantity as the moderating variable, and social trust as the explained variable, to explore the impact of farmer fertility economic value, social value, and emotional value on social trust. Precisely, the direct mechanism, moderating mechanism, and the impact of the children’s economic, social, and emotional values on social trust are given, to reveal the mechanism and effect of fertility value on farmers’ social trust under the moderating effect of farmers’ children quantity.

The second step involved examining and judging the mediating mechanisms and effects of fertility economic value, social value, and emotional value on social trust and focused on testing Hypothesis 2, Hypothesis 3, and Hypothesis 4. Thus, the mediating effect model was constructed (Figure 3) and tested with the farmer fertility economic value, social value, and emotional value as the key explanatory variables, farmers’ social status as the mediating variable, and farmer social trust as the explained variable, to investigate the direct mechanism of the farmer fertility economic value, social value, and emotional value influencing social trust. Moreover, the mechanism and effect of social status on social trust were examined.

In the third step, the mechanism and effect of the farmer fertility economic value, social value, and emotional value influencing social trust under the mediating role of the farmers’ children quantity and their social status were comprehensively considered to mainly test Hypotheses 5 and 6. By constructing a mixed children quantity moderating–social status–mediated effect model (referred to as a mixed moderating mediating effect model; Figure 4), the farmer fertility economic value, social value, and emotional value were taken as the core explanatory variables, the children quantity as the moderating variable, social status as the mediating variable, and social trust as the explained variable. In addition, the model considered the dual mediating and direct moderating effects of the farmers’ children quantity on social status and social trust, as well as the social status–mediated role of the children’s value and children quantity, further elucidating the complex interaction mechanisms and effects among farmer fertility, social status, and social trust.

### 4.2. Model Setting

#### 4.2.1. Moderating Effect Model Setting

According to the empirical strategy, the following moderating effect model was set: (22)$$Trust_{i}=\alpha +\sum_{j=1}^{3}\beta_{1j}^{1}Value_{i}+\sum_{j=1}^{3}\beta_{2j}^{1}Value_{i}\times Quantity_{i}+\beta_{3}^{1}Quantity_{i}+\sum_{k=1}^{1}\delta_{1k}Control_{i}+\mu_{1i}$$

In Model (22), $Trust_{i}$ denotes farmer social trust; $Value_{i}$ denotes farmer fertility value, which is used to determine the direct social trust influence mechanism and impact of children’s economic value, social value, and emotional value; $Quantity_{i}$ denotes the direct social trust effect utility of farmers’ children quantity; $Value_{i}\times Quantity_{i}$ denotes the farmers’ children quantity adjustment mechanism and effect; $Control_{i}$ signifies a series of control variables primarily reflecting farmers’ individual and family endowment information; α, $\beta_{1j}^{1}(j=1,2,3)$, $\beta_{2j}^{1}(j=1,2,3)$, and $\beta_{3}^{1}$, $\delta_{1k}(k=1,2,\cdots l)$ are the parameters to be estimated, giving the direct effect of children’s value, moderating effect of value–social status of children quantity, children quantity direct effect, and control variable effect regression coefficient, respectively; and $\mu_{1i}$ denotes the random disturbance term. The moderating effect model focuses on: (i) the significance of $\beta_{1j}^{1}(j=1,2,3)$. If $\beta_{1j}^{1}(j=1,2,3)$ is significant, then there is a direct social trust impact mechanism and effect on the children’s value; (ii) the significance and strength of $\beta_{2j}^{1}(j=1,2,3)$. If $\beta_{2j}^{1}(j=1,2,3)$ is empirically significant, there is a children quantity moderating mechanism and effect on the farmers’ children’s value, and when $\beta_{2j}^{1}\left(j=1,2,3\right)>0$, the social trust effect of farmers’ children’s value increases with the increase in children quantity. 

#### 4.2.2. Mediating Effect Model Setting

Drawing on Baron and Kenny’s stepwise approach [72], to identify the mediating mechanisms and effects of farmers’ children’s economic value, social value, and emotional value on social trust, the following mediating effect model was set as follows:(23)$$Status_{i}=\alpha +\sum_{j=1}^{3}\beta_{1j}^{2}Value_{i}+\sum_{k=1}^{l}\delta_{2k}Control_{i}+\mu_{2i}$$
(24)$$Trust_{i}=\alpha +\sum_{j=1}^{3}\beta_{1j}^{3}Value_{i}+\beta_{2}^{3}Status_{i}+\sum_{k=1}^{l}\delta_{3k}Control_{i}+\mu_{3i}$$

The mediating effect model has the same variables as the children quantity moderating effect model, except that $Status_{i}$ denotes the social status–mediated variable. The mediating effect model in Equation (23) was used to determine whether there is an effect of farmers’ children’s economic value, social value, and emotional value on the proposed social status–mediated variables; in Equation (24), $\beta_{2}^{3}Status_{i}$ was used to determine the social trust–mediated effect of farmers’ children’s economic value, social value, and emotional value through social status–mediated mechanism, $\sum_{j=1}^{3}\beta_{1j}^{3}Value_{i}$ denotes the direct social trust effect of children’s value, and $\beta_{2}^{3}\sum_{j=1}^{3}\beta_{1j}^{2}Value_{i}$ denotes the social status–mediated effect of children’s value. If the product of regression coefficient $\beta_{1j}^{2}(j=1,2,3)$ and $\beta_{2}^{3}$ is significant, a mediating mechanism and effect of social status exists [64]. Thus, if the regression coefficient $\beta_{1j}^{2}(j=1,2,3)$ is no longer significant, we can judge that there is full mediation; otherwise, it is partial mediation. Notably, if the product term of $\beta_{1j}^{2}(j=1,2,3)$ and $\beta_{2}^{3}$ is of the opposite sign to $\beta_{1j}^{3}(j=1,2,3)$, the mediating effect and the direct effect might cancel each other out and make the social trust effect of farmers’ children’s value insignificant. Furthermore, $\delta_{2k}(k=1,2,\cdots l)$ and $\delta_{3k}(k=1,2,\cdots l)$ are the control variable regression coefficients, and $\mu_{2i}$ and $\mu_{3i}$ are the random perturbation terms. 

#### 4.2.3. Mixed Moderating Mediating Effect Model Setting

Considering the moderating effect of farmers’ children quantity and the mediating effect of social status, the following mixed moderating mediating effect model was constructed to determine the social trust effect of farmers’ children’s economic value, social value, and emotional value: (25)$$Status_{i}=\alpha +\sum_{j=1}^{3}\beta_{1j}^{4}Value_{i}+\sum_{j=1}^{3}\beta_{2j}^{4}Value_{i}\times Quantity_{i}+\beta_{3}^{4}Quantity_{i}+\delta_{4k}Control_{i}+\mu_{4i}$$
(26)$$Trust_{i}=\alpha +\sum_{j=1}^{3}\beta_{1j}^{5}Value_{i}+\beta_{2}^{5}Status_{i}+\sum_{j=1}^{3}\beta_{3j}^{5}Value_{i}\times Quantity_{i}+\beta_{4}^{5}Quantity_{i}+\delta_{5k}Control_{i}+\mu_{5i}$$
(27)$$\begin{array}{l} Trust_{i}=\alpha +\sum_{j=1}^{3}\beta_{1j}^{6}Value_{i}+\beta_{2}^{6}Status_{i}+\sum_{j=1}^{3}\beta_{3j}^{6}Value_{i}\times Quantity_{i}+\beta_{4}^{6}Status_{i}\times Quantity_{i}\\ +\beta_{5}^{6}Quantity_{i}+\delta_{6k}Control_{i}+\mu_{6i} \end{array}$$

Equations (25) and (26) are the children’s value–social status mixed moderation mediating effect models, which are primarily used to assess the relationship of children quantity moderating children’s value–social status and children’s value–social trust and consider the mixed regulatory mediation mechanisms and effect of children’s value on social trust. Equations (25) and (27) are the children’s value–social status–social trust mixed moderation mediating effect models, which are used to determine whether children quantity moderates the children’s value–social trust, children’s value–social status, social status–social trust relations simultaneously to obtain the impact of children’s value on social trust. If the regression coefficient $\beta_{2j}^{4}(j=1,2,3)$ and $\beta_{2}^{5}$ are significant, it indicates that children quantity affects the correlation between farmer fertility value and social trust by moderating the social status–mediated effect in the first half of the path, and by substituting $Status_{i}$ of Equation (25) in Equation (26), we concluded that $\sum_{j=1}^{3}\beta_{1j}^{5}Value_{i}$ is the direct social trust effect of children’s value, $\beta_{2}^{5}\left(\sum_{j=1}^{3}\beta_{1j}^{4}Value_{i}\right)$ is the social status–mediated effect of children’s value, $\beta_{2}^{5}\left(\sum_{j=1}^{3}\beta_{2j}^{4}Value_{i}\times Quantity_{i}\right)$ is the mixed children’s value–social status moderation mediating effect of children’s value, $\sum_{j=1}^{3}\beta_{3j}^{5}Value_{i}\times Quantity_{i}$ is children’s value–social trust direct moderating effect, and $\beta_{2}^{5}\beta_{3}^{4}Quantity_{i}$ is the social status–mediated effect of children quantity. Equations (25) and (27) consider the moderating effect of children quantity on the correlation between farmers’ social status and social trust in Equations (25) and (26). Similarly, substituting Equation (25) into Equation (27), we obtained $\sum_{j=1}^{3}\beta_{1j}^{6}Value_{i}$ as the direct social trust effect on the children’s value, $\beta_{2}^{6}\left(\sum_{j=1}^{3}\beta_{1j}^{4}Value_{i}\right)$ is the social status–mediated effect of children’s value, $\beta_{2}^{6}\left(\sum_{j=1}^{3}\beta_{2j}^{4}Value_{i}\times Quantity_{i}\right)$ is the mixed children’s value–social trust moderation mediating effect of children’s value, $\beta_{2}^{6}\beta_{3}^{4}Quantity_{i}$ is the social status–mediated effect of children quantity, $\sum_{j=1}^{3}\beta_{3j}^{6}Value_{i}\times Quantity_{i}$ is the children’s value–social trust direct moderating effect, $\beta_{4}^{6}\left(\sum_{j=1}^{3}\beta_{1j}^{4}Value_{i}\times Quantity_{i}\right)$ is the mixed social status–social trust moderation mediating effect of children’s value, $\beta_{4}^{6}\left(\sum_{j=1}^{3}\beta_{2j}^{4}Value_{i}\times Quantity_{i}{}^{2}\right)$ is the mixed children quantity–social status–social trust moderation mediating effect of children’s value, $\beta_{4}^{6}\beta_{3}^{4}Quantity_{i}{}^{2}$ is the social status–social trust moderating effect of children quantity and $\beta_{5}^{6}Quantity_{i}$ is the direct social trust effect of children quantity. If the coefficient $\beta_{4}^{6}$ is significant but $\beta_{2}^{6}$ is not, children quantity exerts a moderating effect in the second half of the path of the social status–mediated effect, and by substituting $Status_{i}$ of Equation (25) into Equation (27), the total mixed moderating effect in the first and second half of the path of the mediating effect can be calculated by:$$\beta_{4}^{6}Quantity_{i}\times \left(\sum_{j=1}^{3}\beta_{1j}^{4}Value_{i}+\sum_{j=1}^{3}\beta_{2j}^{4}Value_{i}\times Quantity_{i}\right)$$

If coefficient $\beta_{4}^{6}$ is insignificant but $\beta_{2}^{6}$ is significant, children quantity has a moderating effect in the first half of the social status–mediated effect, and the total mixed moderation mediating effect can be calculated by:$$\beta_{2}^{6}\times \left(\sum_{j=1}^{3}\beta_{1j}^{4}Value+\sum_{j=1}^{3}\beta_{2j}^{4}Value\times Quantity_{i}\right)$$

If the coefficients $\beta_{4}^{6}$ and $\beta_{2}^{6}$ are both significant, children quantity exerts a moderating effect in both the front and back half of the social status–mediated effect, and the total moderation mediating effect can be calculated as follows:$$\left(\beta_{2}^{6}+\beta_{4}^{6}Quantity_{i}\right)\times \left(\sum_{j=1}^{3}\beta_{1j}^{4}Value_{i}+\sum_{j=1}^{3}\beta_{2j}^{4}Value_{i}\times Quantity_{i}\right)$$

### 4.3. Data Sources

In this study, the data were obtained from the China Family Panel Studies (CFPS2018) [73], China Social Science Survey Center, Peking University. The survey covered 31 provinces, municipalities, and autonomous regions (excluding Hong Kong, Macao, and Taiwan) of China and used an implicit stratified multisegment probability sampling technique to track information on fertility, social status, and social trust at the individual, household, and community levels, with a sample size of 44,000 households. The sample is highly representative and provides reliable data support for relevant academic research and public policy analysis. The CFPS 2018 individual, family, and household relationship level data were processed for empirical purposes as follows: (i) farmer was defined as the resident population living in rural areas, and the farmer sample was attained by retaining the individual data of “rural” in the scale question items “Urban–rural classification based on the information of the National Bureau of Statistics”; (ii) the fertility strategy was primarily found in the population of childbearing age, and only the farmer sample aged 15–49 years was retained per the statistical standard of the National Bureau of Statistics for the population of childbearing age. Meanwhile, farmers in this age group were basically constrained by the family planning policy. (iii) After the abovementioned processing, the missing values of key variables were further eliminated, and 6500 valid samples were obtained. 

### 4.4. Definition of Variables and Their Descriptive Statistics

*Explained variables*. The explained variable of the regression equation was farmer social trust. Social trust was derived from people’s psychological trust in other people in society, and it is usually believed that the higher people’s psychological trust in other people in society is, the higher the social trust. The dummy variable “0–1” was constructed using the question, “Do you like to trust or doubt people” in CFPS2018, and respondents who chose “Most people can be trusted” had higher social trust and were assigned a value of 1, whereas respondents who chose “Be as careful as possible” had a lower social trust and were assigned a value of 0.

*Core explanatory variables*. The empirical core explanatory variables were economic, social, and emotional value, which were described by the scale question items “children’s success,” “family succession,” and “family happiness and harmony.” Each item was divided into five options, which were subdivided into five levels from “not important” to “very important” based on their degree and were assigned the values of 1, 2, 3, 4, and 5 (where “1” = not important to the respondent, “5” = very important), and the farmers’ children’s value increases as the value of importance increases.

*Mediator variables*. Farmers’ social status was chosen as the mediator variable, and the question of “own social status in the local area” was chosen directly from the survey scale. The survey scale comprised “social status very low,” “social status low,” “social status average,” “social status high,” and “social status very high,” which were assigned values of 1, 2, 3, 4, and 5, respectively. The higher the value of social status, the higher the farmers’ social status. 

*Moderator variables*. The study used farmers’ children quantity as a moderator variable. As the CFPS2018 scale does not have a direct question item for individual children quantity, the CFPS2018 scale family relationship database provided information on all the children of respondents, which indirectly provided information on the children quantity, thereby determining the children quantity variable. 

*Control variable*. According to the previous theoretical study, farmer fertility, social status, and social trust balance were influenced by macro-factors such as national fertility subsidy, old-age security, and other public services, taxes and prices, as well as by household factors such as individual education, labor ability, employment, income and consumption, and personality factors. Considering the characteristics of the sample data, the control variable was primarily selected from the farmer personality factor and the household factor in the regression. The personal factors control variable mainly included age, gender, marriage, education, health, work, political status, language ability, and social relations of the interviewed farmers. In this study, farmer gender, marital, work, political identity, and language ability variables were derived from the survey scale comprising “respondent gender,” “current marital status,” “current working condition,” “party membership or not,” and “primary language spoken at the interview,”, respectively. If respondents’ gender was “male,” the value was “1”; for “female,” it was “0.” If respondents’ marital status was “married,” the value was “1,” and the other options were “0.” If respondents’ current working condition was “working,” the value was “1,” and the other options were “0.” If respondents’ political status was “party member,” the value was “1,” and the other options were “0.” If respondents’ primary language was “Mandarin” and not “dialect,” then the value was “1”; else, the value was “0.” The age variable mainly controlled the actual age of farmers, and the scale was directly used to fill in the age. The education variable was the number of years of education that farmers had completed. The health variable information was farmers’ five levels of self-rated health, and the questionnaire gave five options of “unhealthy,” “general,” “relatively healthy,” “healthy,” and “very healthy,” which were assigned the values of 1, 2, 3, 4, and 5 in order. In addition, the social relations were obtained from the questionnaire “How good is your relationship with people” and were assigned a value from “0” to “10” depending on the degree of farmer popularity relationship evaluation; the higher the value, the better the farmers’ social relationship. Moreover, the family control variable used two variables of surveyed farmer family economic incomes and family expenditure, taking the logarithm values of households’ “total income in the past 12 months” and “total expenditure in the past 12 months” items in respondents’ answer scale. Table 1 shows the results of descriptive statistics for each variable. 

### 4.5. Multicollinearity Test

Multicollinearity denotes the distortion of model estimates or the instability of regression results in regression models due to highly correlated relationships between explanatory variables. In this study, with reference to the common practice, multicollinearity was tested from both the variance inflation factor (VIF) and tolerance before regression analysis. The larger or smaller the VIF value, the more severe the multicollinearity case. When VIF > 10 or tolerance < 0.1, it indicates the presence of severe multicollinearity, and the regression model should be scientifically adjusted. Table 2 describes the multicollinearity of the regression model. The maximum value of VIF is 1.78, which is much less than 10, and the minimum value of tolerance is 0.562, which is significantly >0.2, suggesting that there is no serious multicollinearity problem in this study, the regression model is more reasonable, and the regression results are credible (Table 2).

### 4.6. Analysis of Regression Results

To improve the estimation accuracy, the interaction terms in the regressions were centralized, and regressions were performed using STATA15.1. The regression results of each model are reported below.

#### 4.6.1. Moderating Effect Model Regression Results 

Table 3 gives the results of the moderating effect model using logit model regression. Model (1) demonstrates that without considering the constraints of control variables, the farmer fertility economic value and social value exerted a negative direct impact on farmers’ social trust, and the children’s emotional value exerted a positive impact on farmers’ social trust. Model (2) demonstrates that without considering the constraints of control variables, the interaction of the farmer fertility economic value with children quantity exerted a significant positive impact on social trust, moderated by the children quantity. The interaction terms of children’s social value and children quantity, children’s emotional value and children quantity, all had significant negative effects on social trust. Comparing Model (1), we found that the direct effect of children’s economic value on peasants’ social trust was no longer significant, the regulatory effect between children’s economic value and the number of children was positive and significant, and the economic value of children enhanced peasant social trust through positive adjustment with the children quantity; this might be due to an increase in the number of peasant children, the decline in the pressure of economic support for a single child of the peasant, and the increase in the absolute income of the children received by peasants, and the relative relaxation of the support game between peasants and children and the increase in social trust. The direct effect and regulatory effect of children’s social value on social trust were negative, and under the role of the regulation of the number of children, the direct negative effect of children’s social value on peasant social trust was reduced. However, when the average number of children of peasants was ≥0.34, the adjustment effect of the number of children enhanced the negative effect of the social value of children on the social trust of farmers, and the more children, the higher the social value of children, and the lower the social trust. Under the negative regulation effect of the number of children, the direct effect of children’s emotional value on social trust was positive, but the effect was reduced, and when the average number of children of farmers was >2.235, the increase in children’s emotional value inhibited the development of peasant social trust as a whole, and might have increased the number of children, which might further complicate the relationship between farmers and their children, intensify the game between children, and reduce social trust. Model (3) is based on Model (2) with the addition of control variables. The coefficient size and significance of the key variables in this model aligned with Model (2), except that the direct effects and moderating effects of children’s economic value, social value, and emotional value were both reduced, which adequately proves the robustness of children’s value on farmers’ social trust regression results. Models (1)–(3) show that the farmers’ children’s economic, social, and emotional values exerted asymmetric direct effects on farmers’ social trust and children quantity moderating effects. Owing to the asymmetric moderating effect of children quantity, the impact of children’s economic value, social value, and emotional value on farmers’ social trust was asymmetric, and the final effect of farmers’ children’s value on farmer social trust was determined by the direct impact of each child’s value and children quantity moderating effect. This provides empirical support for the hypothesis that the farmer fertility value affects social trust and validates the reliability of the partial content of Hypotheses 1, 5, and 6.

#### 4.6.2. Mediating Effect Model Regression Results 

Table 4 shows regression results by the mediating effect model. According to Model (4), the regression coefficient of children’s economic value on social status was not significant, and children’s social value and emotional value exerted significant positive effects on social status, indicating that children’s economic value had a neutral effect on farmers’ social status, but children’s social value and emotional value exerted a direct positive impact on farmers’ social status, and the growth of children’s social value and emotional value enhanced farmers’ social status. This is consistent with the expectations of Hypothesis 2, proving its reliability and corroborating the results of Luo et al. [4] on the relationship between farmers’ social status and fertility choices, although it differs in that the impact of childbirth on farmers’ social status was not attained through the economic value mechanism through social value and emotional mechanisms.

In Model (5), there was no direct effect of children’s economic value on social trust; the direct effect of children’s social value on farmers’ social trust was significantly negative, while a significantly positive and direct effect of emotional value was found. Thus, the net direct effect of children on social trust depends on the combination and the effect size of children’s social value and emotional value. When the social value negative direct effect was greater than the positive direct effect of emotional value, the increase in the children’s value directly reduced farmer social trust; otherwise, it would increase farmers’ social trust; this is consistent with the model results in Table 2, further validating Hypothesis 1.

The social status coefficient in Model (5) was significantly positive, while economic value exerted a negative complete social status–mediated effect and inhibited social trust development. Moreover, social value and emotional value were still significant, indicating that social status is a noncomplete mediator indicating that farmers’ children’s social value and emotional value affect social trust. Considering that social status and social trust are nonequivalent scale variables and the regression coefficient obtained was not comparable, the ratio of the fertility value regression coefficient to its standard error in Model (4) and the ratio of the social status regression coefficient to its standard error in Model (5) were calculated using Iacobucci’s method, using $Z_{a}=Value/SE(Value)$ and $Z_{b}=Status/SE(Status)$, respectively, to make the regression results comparable, and the size of the mediating effect was obtained by the calculation of $Z_{a}\times Z_{b}$ [74]. Then, the significance of the mediating effect was determined by the RMediation product fractional-step method in the R software. If the test confidence interval did not contain 0, it indicated that the mediating effect is significant [75]. The test revealed that the social status–mediated effect size of farmers’ children’s social value acting on social trust was (0.116/0.014) × (0.08/0.026) = 25.495, which is statistically significant. The social status–mediated effect size of farmers’ children’s emotional value on social trust was (0.062/0.02) × (0.08/0.026) = 9.539, and the social status–mediated effect of emotional value on social trust was also significant. Furthermore, the children’s emotional value positively influenced the development of farmer social trust through direct and social status–mediated mechanisms. The full mediated utility of social status for children’s economic value was –2.105 and significant. Children’s social value positive mediating effect was the largest, the positive mediating effect of emotion was at an intermediate level, and the negative mediating effect of economic value was the smallest.

According to Table 4, farmers’ children’s value direct mechanism and social status mediation mechanism asymmetrically affect farmer social trust. 

The farmers’ children’s economic value exerted no direct impact on farmers’ social trust, only a negative full social status–mediated effect. The negative full mediating effect of social status, in which children’s economic value was only negative, can inhibit the social trust development. Children’s social value exerted a negative direct effect and a positive social status–mediating effect on farmers’ social trust, and the social status–mediated effect of children’s social value was smaller than the direct effect, and the growth of children’s social value inhibited the farmers’ social trust development. The direct impact of children’s emotional value and social status–mediated effect were both positive, but the direct effect was greater than the mediating effect, so the direct effect mechanism of children’s emotional value and social status–mediated mechanism asymmetrically contributed to the farmers’ social trust development, mainly through direct effects to enhance the social trust of farmers. As the full negative mediating effect of the farmers’ children’s economic value on farmers’ social trust was much smaller than the positive mediating effect of the children’s social and emotional value, the farmers’ social status and social trust were higher. The regression results validated the reasonableness of Hypotheses 2–4 well.

#### 4.6.3. Mixed Moderation Mediating Effect Model Regression Results 

Table 5 shows the regression results of two mixed moderation mediating models. The regression information of the mixed children’s value–social status–mediated effect model comprised Models (6) and (7) showing that: (i) there was no significant direct effect of children’s economic value on farmers’ social status, and there was a positive direct effect of children’s social value and emotional value on farmers’ social status. With the reform and opening up of China and the one-child policy, farmer economic independence is growing on the one hand, and the income level is increasing rapidly on the other. The social status effect was significantly decreased in children’s economic value and significantly increased in children’s economic value and emotional value, and regression results are reflected this trend. (ii) No moderating effect was found on the correlation between farmers’ children’s economic value–social status and children’s emotional value–social status in children quantity statistics. A negative moderating effect was observed on the correlation between children’s social value–social status, suppressing the social status effect of social value. This could be because China’s long-term strict family planning policy and the urbanization, industrialization, and marketization development, fertility cost growth, and children quantity growth is more reflected in the clan village social status value generated by the succession, which in turn reinforces male preference and family’s male marriage competition pressure, weakening farmers’ social status. The social status effect of economic value and emotional value was not significant because it is written off by the children quantity factor. In addition, children quantity exerted a negative moderating effect on the first half of the social status–mediated pathway, indicating that although the academic community recognizes a positive reinforcing relationship of “the richer, the more children” between farmers’ social status and children quantity, there is also a reverse weakening relationship of “the poorer, the more children” between social status and children quantity. These two evolutionary relationships were attained through the children’s social value and emotional value direct effects and the children quantity regulation mechanism of social value. (iii) The direct social trust effect of farmers’ children’s economic value was not significant; children’s social value exerted a significant positive direct social trust effect, and children’s emotional value exerted a significant negative direct social trust effect. (iv) Farmers’ social status would enhance their social trust because farmers with a higher social status have more resources for developing trusting relationships and a relatively higher level of social trust [51]. (v) Combined with Model (6), a significant social status–mediated effect of children’s economic value was noted on farmers’ social trust and children’s social value, and emotional value had a significant positive social status–mediated effect on farmer social trust and promoted farmers’ social trust development through a social status–mediated effect, while the economic, social, and emotional values of children promoted the development of peasants’ social trust through the intermediary role of social status. (vi) The economic value of children had a negative completely mixed adjustment mediation effect on social trust, with a size of 0.079 × (−0.012$Quantity_{i}$), which inhibited the growth of peasant social trust. The social value of children had a significant negative mixed regulation mediation effect on the social trust of farmers, and the mixed regulation mediation effect was 0.079 × (−0.031$Quantity_{i}$). Emotional value had a completely positive mixed–regulation mediating effect, with a size of 0.079 × (0.022). Although the mixed adjustment mediation effect of children’s emotional value was positive, the mixed adjustment mediation effect of the economic value of peasant children’s birth and social value was negative, and the net effect of the total child’s value–social status mixed adjustment intermediary mechanism was 0.079 × (−0.021$Quantity_{i}$), while the social value of the child’s reproductive value mixed regulation intermediary role as a whole inhibited the growth of peasant social trust; the more children, the greater the inhibition effect of the mixed adjustment effect of the child’s social value on the social trust of the peasant year. This could be because more children leads to the reduction in the peasant social trust resource input, the increase in social survival competition pressure, and the greater marginal reduction in social trust. (vii) The children quantity moderation regression coefficients of economic value, social value, and emotional value were 0.100, −0.045, and −0. 079, respectively, which are statistically significant, suggesting that the effect of the economic value moderating mechanism was positive, while the social value moderating effect and emotional value moderating effect were both negative. (viii)The number of children had a completely negative social status mediation effect but no direct effect, the number of children inhibited social trust through social status mediation, the direct mechanism did not affect social trust, and there was no social status–social trust adjustment effect. Hence, Hypotheses 4,5, and 6 were verified.

Models (6) and (8) in Table 5 present the regression information of the mixed children’s value–social status–social trust moderation mediating effect model. Compared with Model (6), Model (7) only had one more validation item about children quantity regulating social status–social relations because the social status and children quantity cross-term coefficient of regression model was not significant, and children quantity did not regulate the social status–social trust relationship, so the children’s value’s social status–social trust mixed moderation mediating effect, the children quantity–social status–social trust mixed moderation mediating effect of children’s value, and the children quantity social status–social trust moderating effect were not significant, while the regression coefficient of other effects aligned with the children’s value–social status mixed–moderated mediating effect model regression situation of Models (6) and (7), with the same effect coefficient size and action direction, suggesting that children quantity did not have a moderating effect on the second half of the social status–mediated path, proving that the regression results were robust. This further supports Hypotheses 5 and 6 and also verifies the reliability of other hypotheses. To simplify the discussion, only the mechanism and transmission path of children’s value affecting social trust are given here (Figure 5).

In addition, as shown in Table 2, Table 3 and Table 4, the impact of the control variables on the social status and social trust of farmers is relatively stable and consistent. The older the farmer, the higher the social status and social trust. Male gender degraded male social status but did not affect social trust, which could be related to the growth of women’s status in China, gender imbalance, and fierce marital competition. Marriage exerted no significant impact on the social status and social trust of farmers. Education inhibited the improvement of peasants’ social status, but enhanced their social trust. The higher the level of health, the higher the level of social status and social trust of farmers. Employment did not affect the social status of the farmer’s year, but employment could enhance their social trust. Party membership not only significantly enhanced the social status of peasants but also markedly improved the level of social trust of peasants, and the marginal effect was far greater than that of other control variables; this is in line with China’s reality because in the Chinese peasant class, party members are usually elites, have a relatively high social status, and have political resources that ordinary farmers do not have, personal development is relatively more successful than ordinary farmers, and social trust is naturally higher. Speaking Mandarin will inhibit the social status of peasants, but it will enhance the social trust of peasants; this may be related to the localization of the peasant social interaction circle, as the peasant social circle is relatively closed, speaking Mandarin without speaking a dialect is regarded as out of place and discriminated against by insiders, and the social status in the circle declines owing to discrimination, but peasants speaking Putonghua can expand exchanges, broaden their horizons, and eliminate language discrimination, thereby decreasing communication barriers and augmenting social trust. Social relations simultaneously enhance the social status and social trust of farmers. Family income has improved the social status of farmers, but the coefficient of influence on the social trust of farmers is negative, and it is not statistically significant. Furthermore, the impact of peasant household expenditure on farmers’ social status and social trust is not significant.

## 5. Conclusions

With the development of China’s economic and social reform and opening up, asymmetric changes have been witnessed in the choices of farmers’ life and employment, fertility, social status, and social trust competition, and the choices of fertility, social status, and social trust combinations are bound to evolve dynamically; however, the combination of child fertility, social status, and social trust choices is balanced, and the intrinsic decision mechanism and role utility are issues. There is not a fully established unified framework for systematic theoretical and empirical research, and most of the existing studies are based on the consumption–child quantity–quality selection theory of Becker and Lewis [18], discussing the equilibrium of consumption, childbirth quantity–quality selection, and elucidating the economic value of child consumption or investment in the quantitative–quality choice of children [20,21,22]. This study extended the existing consumption–children quantity–quality choice model to build a theoretical and empirical model of farmer consumption–fertility–status–trust social choice, and draws the following conclusions: (i) farmers’ rational consumption choices based on the comparative advantage mechanism, fertility, social status and social trust equilibrium combination, fertility, social status, and social trust choices are only crucial options of farmers’ social behavior choice equilibrium and have mutual competition substitution. Our findings demonstrate that farmers not only make quantitative-value choices [30] but also optimize choices for a combination of childbearing decision–social status and social trust; (ii) farmer fertility pursues the children’s emotional value, social value, and economic value, and balances the rational selected fertility number, emotional value, social value, and economic value based on the comparative advantage, the quantitative–quality selection of childbirths in the theories such as those of Becker and Lewis [18] and Yuan [23], which are actually in line with the number of births–economic value selection in this study; thus, the number of children–economic value can explain the number of children born to quality behaviors discussed by Becker and Lewis [18], Liu and Lu [21], and Yuan [23]. Meanwhile, it could also be from the number of children born—social value. The number of childbirths–emotional value dimension provides a richer but more unique explanation of farmers’ fertility behavior [18,20,22]. It is precisely because of the multidimensional value orientation of the choice of the number of children of peasants, as well as the competitive substitution of social status and social trust, which has led to the transformation of farmers’ fertility quantity preferences to quality preferences since China’s reform and opening up [21], and the increasing cost of farmers to obtain the value of children’s birth and the growing demand for social status and social trust have accelerated the development of farmers with fewer children [6]. (iii) The child-giving behavior affects the social status and social trust of farmers asymmetrically through the direct mechanisms, regulatory mechanisms, social status mediation mechanisms, and mixed adjustment mediation mechanisms of the number of children, emotional value, social value, and economic value, rather than the simple number of children born—social status balance as believed by other researchers [4,27], nor will it be the simple balance of economic status and social trust as proposed by Shen and Zhang [8] or the intergenerational flow of social trust and social status Shan et al. [76] suggest, which also includes the moderating role, mediating role, or even the mixed regulation mediation role of the number of births and multidimensional values [8,77], and together with the individual education, health, politics, social relations, employment, and other factors of farmers [5,51,60,77].

Empirical studies have reported that the economic value of child fertility has no direct effect on social status or the moderating effect of child number, and no positive correlation exists between the economic value of child fertility and social status of farmers [4,13,42,45], which Utomo et al. found as well [30] for Chinese farmers with a low social status and a rapid decline in the number of children. In fact, according to the empirical results, the fertility of peasant children enhances their social status through social and emotional values, rather than the economic value of childbirth. Meanwhile, the birth of peasant children only has a negative social value–social status adjustment effect of the number of children, and there is no significant economic value of children—social status and children’s emotional value—social status adjustment effect. As the number of children exceeds 3.8, the social value of peasant children inhibits the growth of their social status, and the more children there are, the greater the social status inhibition effect of social value, which even writes off the status contribution of children’s emotional value, resulting in children’s fertility inhibiting the growth of social status. Shi demonstrated a negative correlation between fertility and status [57]. Thus, although the direct impact of the number of peasant children on social status is not significant, the correlation between the number of children born and social status is not a simple positive correlation, negative correlation, or definite relationship [4,27,42,57], nor does the negative correlation weaken or disappear when fertility is lower than the replacement level [30], but it depends on the combination of changes between children’s social value, emotional value, and the number of children, and is also affected by the economics of peasant families, personal political identity, social relations, health, education, age, and constraints due to factors such as gender.

Further empirical evidence also revealed that the impact of peasant childbearing and social status on social trust is particularly complex. First, the economic value of peasant children exerts no direct impact on social trust and will not directly affect peasant social trust. Nevertheless, the economic value of children increases the level of social trust through the adjustment mechanism of the number of children, and the growth of social trust is inhibited by the intermediary mechanism of social status and the mixed adjustment mechanism of economic value–social status of the number of children. Regardless of whether the growth of the economic value of peasant children promotes social trust, which is also regulated by the number of children, when the number of children is <3.135, the social trust of peasants declines with the value of children, and when the number of children is >3.135, it increases with the economic value of children. Currently, if the average number of children born to peasants is much lower than 3.135, the growth of the economic value of children inhibits the social trust of farmers. This leads to the increase in household income caused by the increase in the value of farmers themselves and their children in the process of economic and social development, and the economic dependence of farmers on relatives and friends is reduced, which also decreases the support for relatives and friends, thereby undermining social trust [78,79]. Second, the social value of children exerts a negative direct effect on social trust, a positive social status mediation effect, a negative number of children needed to regulate the effect of regulation, and a negative mixed adjustment mediation effect. Regardless of how the combination of children’s social value and the number of children changes, the growth of children’s social value will inhibit the growth of peasant social status, and the more children, the greater the social value, and the greater the inhibition effect. Preliminary calculations demonstrated that the net effect of social trust impact produced by the four mechanisms of the social value of peasant children is −(0.1353+0.0475n)S, showing that despite the number of peasant children, the increase in peasant social value will decrease social trust. With this result, Fukayama [54] and Xiang and Li [77] claimed that Chinese farmers prefer blood-led social trust preferences; the higher the social value of children, the more farmers attach importance to the social trust of blood-related children, and the smaller the trust dependence on society, the lower the social trust. Hence, the higher the number of children, the greater the total social trust inhibition effect of the growth of the social value of peasant children. Third, the emotional value of children exerts a positive direct impact on the social trust of farmers, a positive social status mediation effect, positive social status—the number of children needed to regulate the mediation effect, and a negative number of children; the total impact effect is that when the number of children is <1.347, the growth of children’s emotional value will enhance the level of social trust of farmers; this may be because the number of children exceeds the threshold, the feelings of family members are too complex, the competition for children’s emotional value tends to be fierce, or because the peasants have a lower social status, limited resources, and limited ability to invest in their children’s emotional value, so social trust will be lower [5,44,45,53]. Fourth, the number of children has no direct or intermediary effect on social status; it has a positive regulatory effect on children’s emotional value—social status, economic value—social trust, but a negative regulatory effect on children’s social value—social status and emotional value—social trust; it has no significant regulatory effect on economic value social status, social value social trust, social status social trust, the direct effect of social trust, and intermediary effect of social status, and the correlation between the number of farmers’ children, social status, and social trust is much more complex than that envisaged by Luo et al. [4] and Shan et al. [76]. Fifth, social status exerts a positive and direct effect on social trust and does not symmetrically transmit the mediating effect of children’s value, the quantitative adjustment effect of children’s value, and the mixed adjustment of social value–social status, but there is no social trust mediation effect of the number of children. This indicates that farmer fertility in China affects social status and social trust primarily through children’s value rather than children quantity. Children quantity only plays a moderating role, and the influence of a children’s value is primarily due to children’s emotional value and social value mechanisms. Besides, the effects of social status and social trust of children’s economic value decline. The effect of farmers’ social status on social trust exerts both a direct role and an asymmetric children’s value mediating role and a mixed regulating mediation role in children quantity. In addition, in the process of influencing social trust through the combination of complex mechanisms, the number of peasant children’ births, the value and status of peasants’ children, and many control factors, such as individual farmers and families, jointly affect the social trust of farmers.

This study argues that with the rapid growth of economy and society, the choice of peasant children’s child reproduction, social status, and the social trust game is gradually becoming biased toward social status competition and social trust development, and the value of child fertility is constantly strengthened, and the choice of the quantity–quality of childbirth is actually the choice of the quantity and value of children’s birth, which exerts a complex impact on the social status and social trust of farmers. The sustained growth of peasant incomes and the increase in the level of social public security, the reduction in the economic value of peasant children’s birth, and the significant increase in the demand for social value and emotional value will inevitably lead to the adjustment of the number of children and changes in the structure of reproductive value, which, in turn, will affect the development of peasants’ social status and social trust. Besides liberalizing the number of children born and increasing birth subsidies, the government should also start from the choice of optimizing the birth of peasant children’s children–social status–social trust, optimize government policies, focus on improving the emotional and social value of children, promote social status mobility, enhance the ability of comprehensive social management, break the dilemma of peasant social status locking, and promote the development of peasant social trust.

## Figures and Tables

**Figure 1 ijerph-19-04759-f001:**
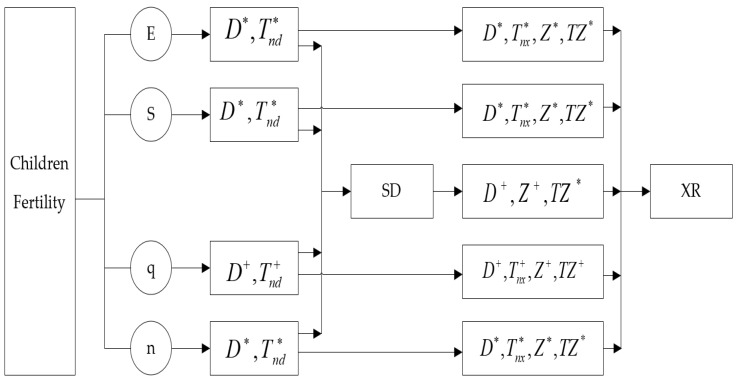
Theoretical mechanisms and effects of fertility, social status, and farmer social trust influence. Note: (1) E, S, and q denote the economic, social, and emotional values of children’s fertility; *n* represents the number of children, SD represents social status, and XR represents social trust; (2) D stands for direct mechanism; *T_nd_* represents the child’s value–social status adjustment mechanism, *T_nx_* represents the child’s value–social trust adjustment mechanism, Z represents the intermediary mechanism, and TZ represents the mixed adjustment intermediary mechanism; (3) “*” indicates that the direction of action of the mechanism depends on the combination of parameters, and the direction is uncertain, and “+” indicates positive effect and “–” indicates negative effect.

**Figure 2 ijerph-19-04759-f002:**
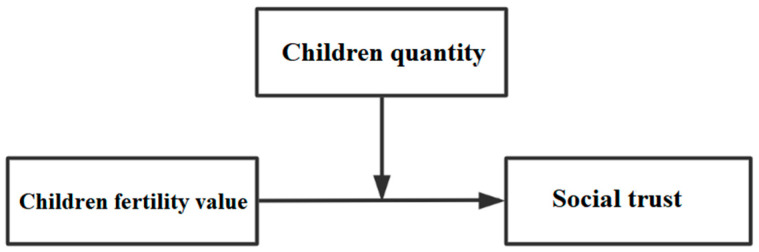
The children quantity moderating effect model of fertility value and social trust.

**Figure 3 ijerph-19-04759-f003:**
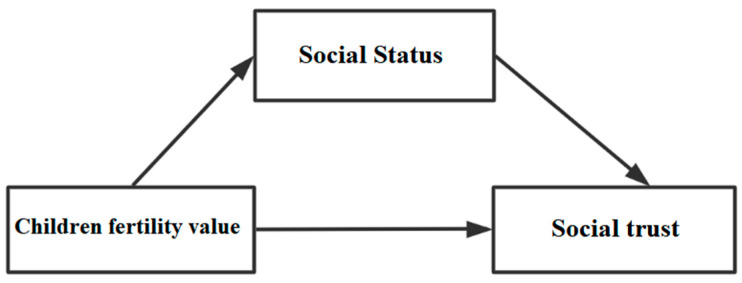
Social status–mediated effect model of fertility value influencing social trust.

**Figure 4 ijerph-19-04759-f004:**
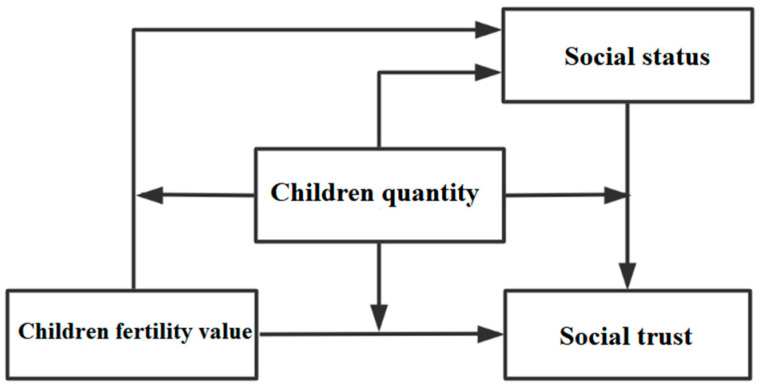
Children quantity–social status mixed moderation mediating effect model of fertility influencing social trust.

**Figure 5 ijerph-19-04759-f005:**
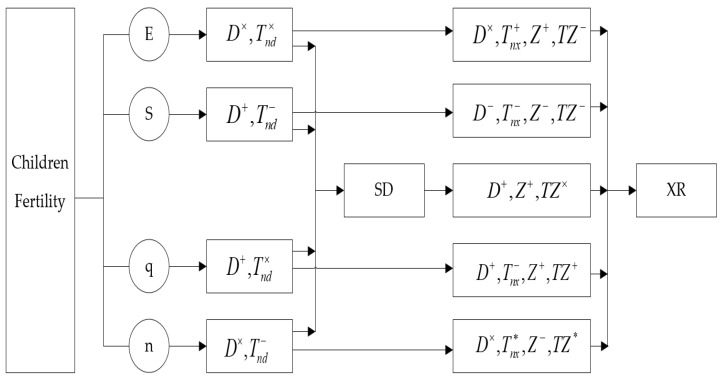
The actual mechanism and transmission path of children’s value influencing farmers’ social trust. Note: (1) E, S, and q denote the economic, social, and emotional values of children’s fertility; *n* represents the number of children, SD represents social status, and XR represents social trust; (2) D signifies direct mechanism, *T_nd_* represents the child’s value–social status adjustment mechanism, *T_nx_* represents the child’s value–social trust adjustment mechanism, Z denotes the intermediary mechanism, and TZ indicates the mixed adjustment intermediary mechanism; (3) “*” indicates that the direction of action of the mechanism depends on the combination of parameters, and if the direction is uncertain, “+” indicates positive effect, “−” indicates negative effect and “×” indicates that the empirical test has no significant effect.

**Table 1 ijerph-19-04759-t001:** Descriptive statistics of the variables.

Variable Name		Sample Size	Average Value	Standard Deviation	Minimum Value	Maximum Value
Explained variable	Social trust	6500	0.533	0.499	0	1
Core explanatory variables	Economic value	6500	4.550	0.814	1	5
	Social value	6500	4.154	1.088	1	5
	Emotional value	6500	4.660	0.716	1	5
Mediator variable	Social status	6500	3.013	1.055	1	5
Moderator variable	Children quantity	6500	1.514	1.011	0	7
Control variable	Age	6500	35.873	8.848	16	49
	Gender	6500	0.494	0.500	0	1
	Marital status	6500	0.824	0.381	0	1
	Education status	6500	7.773	4.422	0	19
	Health condition	6500	3.249	1.177	1	5
	Working condition	6500	0.807	0.395	0	1
	Political identity	6500	0.052	0.221	0	1
	Language ability	6500	0.575	0.495	0	1
	Social relations	6500	7.022	1.956	0	10
	Family income logarithm	6500	10.529	1.183	0	14.286
	Family expenditure logarithm	6500	10.183	1.071	0	14.221

**Table 2 ijerph-19-04759-t002:** Regression model multicollinearity.

Variable	Multicollinearity Statistics	
	Variance Inflation Factor (VIF)	Tolerance
Economic value	1.49	0.673
Social value	1.41	0.711
Emotional value	1.33	0.752
Economic value × children quantity	1.54	0.649
Social value × children quantity	1.45	0.687
Emotional value × children quantity	1.37	0.728
Children quantity	1.78	0.562
Social status × children quantity	1.08	0.927
Social status	1.17	0.857
Age	1.63	0.613
Gender	1.15	0.869
Marital status	1.65	0.606
Education status	1.52	0.658
Health condition	1.11	0.899
Working condition	1.11	0.903
Political identity	1.04	0.964
Language ability	1.13	0.885
Social relations	1.09	0.914
Family income logarithm	1.27	0.785
Family expenditure logarithm	1.23	0.812

**Table 3 ijerph-19-04759-t003:** Regression results of farmers’ children’s value influencing the social trust moderating effect model.

Variable	Model (1)	Model (2)	Model (3)
Economic value	–0.063 * (0.0370)	–0.039 (0.038)	–0.045 (0.039)
Social value	–0.142 *** (0.027)	–0.124 *** (0.028)	–0.116 *** (0.028)
Emotional value	0.187 *** (0.039)	0.179 *** (0.040)	0.104 ** (0.042)
Economic value × children quantity		0.102 *** (0.036)	0.099 *** (0.036)
Social value × children quantity		–0.053 ** (0.027)	–0.048 * (0.028)
Emotional value × children quantity		–0.080 ** (0.036)	–0.077 ** (0.037)
Children quantity		–0.111 *** (0.025)	–0.043 (0.034)
Age			0.007 * (0.004)
Gender			0.062 (0.054)
Marital status			0.053 (0.086)
Education status			0.059 *** (0.007)
Health condition			0.110 *** (0.023)
Working condition			0.113 * (0.068)
Political identity			0.292 ** (0.120)
Language ability			0.111 ** (0.054)
Social relations			0.056 *** (0.014)
Family income logarithm			–0.001 (0.024)
Family expenditure logarithm			0.028 (0.027)
Constant term	0.138 (0.184)	0.155 (0.190)	–1.578 *** (0.369)
Sample size	6500		

Note: *, **, and *** indicate that the coefficients are significant at the 10%, 5%, and 1% levels, respectively; the values in parentheses below the coefficients are their standard errors.

**Table 4 ijerph-19-04759-t004:** Mediating effect model regression results of farmers’ children’s value influencing social trust.

Variable	Farmers’ Social Status	Farmer Social Trust
	Model (4)	Model (5)
Economic value	–0.013 (0.019)	–0.050 (0.0381)
Social value	0.116 *** (0.014)	–0.136 *** (0.028)
Emotional value	0.062 *** (0.020)	0.094 ** (0.041)
Social status		0.080 *** (0.026)
Age	0.013 *** (0.002)	0.005 (0.004)
Gender	–0.072 *** (0.027)	0.067 (0.054)
Marital status	0.045 (0.037)	–0.005 (0.0759)
Education status	–0.013 *** (0.004)	0.062 *** (0.007)
Health condition	0.116 *** (0.012)	0.101 *** (0.023)
Working condition	–0.048 (0.034)	0.119 * (0.068)
Political identity	0.183 *** (0.053)	0.282 ** (0.119)
Language ability	–0.061 ** (0.027)	0.125 ** (0.054)
Social relations	0.098 *** (0.008)	0.049 *** (0.014)
Family income logarithm	0.037 *** (0.010)	–0.007 (0.024)
Family expenditure logarithm	–0.021 (0.014)	0.030 (0.027)
Constant term	0.771 *** (0.179)	–1.510 *** (0.368)
Sample size	6500	

Note: *, **, *** indicate that the coefficients are significant at the 10%, 5%, and 1% levels, respectively; the values in parentheses below the coefficients are their standard errors.

**Table 5 ijerph-19-04759-t005:** Mixed moderation mediating effect model regression results of farmers’ children’s value influencing social trust.

Variables	Social Status	Social Trust	Social Trust
	Model (6)	Model (7)	Model (8)
Economic value	−0.021 (0.019)	–0.044 (0.039)	–0.043 (0.039)
Social value	0.118 *** (0.014)	–0.126 *** (0.028)	–0.125 *** (0.028)
Emotional value	0.065 *** (0.021)	0.099 ** (0.042)	0.098 ** (0.041)
Economic value × children quantity	–0.012 (0.017)	0.100 *** (0.036)	0.101 *** (0.036)
Social value × children quantity	–0.031 ** (0.014)	–0.045 * (0.028)	–0.048 * (0.027)
Emotional value × children quantity	0.022 (0.017)	–0.079 ** (0.038)	–0.079 ** (0.037)
Children quantity	–0.008 (0.017)	–0.042 (0.034)	–0.042 (0.034)
Social status × children quantity			0.015 (0.025)
Social status		0.079 *** (0.026)	0.076 *** (0.026)
Age	0.013 *** (0.002)	0.006 * (0.004)	0.006 * (0.004)
Gender	–0.078 *** (0.027)	0.068 (0.055)	0.068 (0.055)
Marital status	0.043 (0.043)	0.05 (0.086)	0.053 (0.086)
Education status	–0.013 *** (0.004)	0.06 *** (0.007)	0.06 *** (0.007)
Health condition	0.116 *** (0.012)	0.101 *** (0.023)	0.101 *** (0.023)
Working condition	–0.047 (0.034)	0.118 * (0.068)	0.117 * (0.068)
Political identity	0.185 *** (0.053)	0.278 ** (0.120)	0.277 ** (0.121)
Language ability	–0.066 ** (0.027)	0.116 ** (0.054)	0.116 ** (0.054)
Social relations	0.098 *** (0.008)	0.049 *** (0.014)	0.049 *** (0.014)
Family income logarithm	0.038 *** (0.01)	–0.004 (0.024)	–0.004 (0.024)
Family expenditure logarithm	–0.021 (0.014)	0.029 (0.027)	0.029 (0.026)
Constant term	0.787 *** (0.180)	–1.643 *** (0.370)	–1.646 *** (0.367)
Sample size	6500		

Note: *, **, *** indicate that the coefficients are significant at the 10%, 5%, and 1% levels, respectively; the values in parentheses below the coefficients are their standard errors.

## Data Availability

The data of CFPS2018 is publicly available.

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
