# Peer review of "Development of Fertility, Social Status, and Social Trust of Farmers"

_ijerph, 2022, doi:10.3390/ijerph19084759_

Round 1

Reviewer 1 Report

The article aims to study the complex causal relationships between fertility, social status and social trust of Chinese farmers. The main focus is on fertility, i.e. the quantity, emotion value, social value and economic value of children.

Its main contributions are to show that the most important factor of social status and social trust is children's value and no children's quantity. In that sens the best way to improve farmer's social status and social trust is to increase the children's emotional and social value.

The manuscript is quite clear, but some sentences/paragraphs could be shortened in order to make them more clear and understandable for the reader. 

Some concepts could also be more distinguished (for example: moderato and mediator)

The design is also clear and relevant to the thematic. 

Concerning the data, it is not clear if it is possible to have access to these data or not. No link or specific reference are given in the bibliography to have access to the main source of data: China Family Panel Studies. Therefore, when the authors say that these data are "reliable", we have to take their words for it, but we have no proof they are.

Reviewer 2 Report

Thank you for the opportunity to review this article. After reading it, I found the following aspects related to:

1. Abstract. The abstract does not clearly state the purpose or objectives of the research conducted by the author. I suggest the author to do this and add the results and conclusions of his research.

2. Introduction. As in the case of the abstract, the author does not clearly state the purpose or objectives of his research. The introduction is confused with the part of the specialized literature. I suggest the author to respect the scientific structure of a specialized article: introduction, specialized literature, research methodology, analysis of the obtained results and their interpretation, conclusions.

3. Theoretical models. This section is described only in terms of research hypotheses not being argued based on shortcomings in the literature. I suggest the author to rename this section Literature and to treat as such the part related to research hypotheses.

4. Empirical study. This section should be dedicated to Research Methodology. The author does not describe the research method used but only tests the research hypotheses. I suggest the author clarify these issues. After this section there should be a separate section called Analysis and interpretation of the results obtained, but it seems to be missing!

5. Conclusions. In this section the author describes more the results of the study and does not present the implications brought by his research, does not emphasize his contribution in the field and does not present the limits of his study. I suggest the author to do this! 

Reviewer 3 Report

  1. A conceptual framework should be diagrammatized and explained before the models.
  2. Some of hypotheses are confusing, and they should be mutually exclusive. Hypotheses 1 and 3 should be broken down into two for clarity. 
  3. Simply the 'theoretical mechanism' or I rather call it 'hypothetical mechanism'. Details of what should be done is in the manuscript.
  4. Interaction terms should be determined based on diagnostic test to construct correlation matrix with condition index, variance inflation factor (VIF), eigenvalue to suggest or guide predictors from which interaction terms will be constructed irrespective of available literature.
  5. There is no discussion section in the manuscript. This is a major gap that need to be filled. You may convert the conclusion section into discussion section and support your key findings with relevant literature. 
  6. Rewrite your conclusion section to reflect key implications for farmers and the government based on your findings.
  7. Other minor comments are in the body of the manuscript. 

Round 2

Reviewer 2 Report

Thanks to the authors for the suggestions they made and fulfilled!